

# 3D gravity in a box

**Per Kraus, Ruben Monten and Richard M. Myers**

Mani L. Bhaumik Institute for Theoretical Physics, Department of Physics & Astronomy,
University of California, Los Angeles, CA 90095, USA

## Abstract

The quantization of pure 3D gravity with Dirichlet boundary conditions on a finite boundary is of interest both as a model of quantum gravity in which one can compute quantities which are "more local" than S-matrices or asymptotic boundary correlators, and for its proposed holographic duality to $T\overline{T}$-deformed CFTs. In this work we apply covariant phase space methods to deduce the Poisson bracket algebra of boundary observables. The result is a one-parameter nonlinear deformation of the usual Virasoro algebra of asymptotically AdS$_3$ gravity. This algebra should be obeyed by the stress tensor in any $T\overline{T}$-deformed holographic CFT. We next initiate quantization of this system within the general framework of coadjoint orbits, obtaining — in perturbation theory — a deformed version of the Alekseev-Shatashvili symplectic form and its associated geometric action. The resulting energy spectrum is consistent with the expected spectrum of $T\overline{T}$-deformed theories, although we only carry out the explicit comparison to $\mathcal{O}(1/\sqrt{c})$ in the $1/c$ expansion.

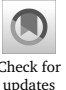

# 1   Introduction

Within the framework of our current understanding of quantum gravity, the only observables with a mathematically precise definition involve asymptotic quantities, such as the S-matrix in Minkowski space or boundary correlators in (A)dS. Yet in order to understand a variety of problems of conceptual and observational interest, notably those involving black holes and cosmology, it seems necessary to broaden the range of calculable quantities, and indeed much effort has gone into trying to define observables which are more "local" in character; some references, mainly within the AdS/CFT context, include [1–14].

By obvious analogy with the case of the electromagnetic field, it is natural to ask whether it is sensible to define quantum gravity in a "box", in the sense of imposing boundary conditions on the metric on a timelike boundary of finite spatial extent. At the classical level, such a setup is relevant for numerical relativity; for a review see [15]. Again at the classical level, this has been considered [16] within the framework of the fluid-gravity correspondence [17], where one considers the boundary to enclose a black hole whose horizon leads to dissipative effects. The problem of making sense of quantum gravity confined to a subregion is of interest in the context of entanglement; e.g. [18]. At the level of free field theory, the gravitational analog of the Casimir effect between parallel plates was calculated in [19]. Various subtleties involving Dirichlet boundary conditions can arise, such as apparent superluminal propagation [20]. Also, in the case of Euclidean signature, the Einstein equations with Dirichlet boundary conditions have the unpleasant feature of not admitting a sensible perturbation theory; see [21] for a review. A separate type of question is whether Dirichlet boundary conditions on the metric are physically realizable — is it possible to construct the gravitational analog of a conducting plate? Other references with a similar spirit as the present work include [22–27].

The situation is simpler in lower dimensional gravity due to the absence of local degrees of freedom. In two-dimensional Jackiw-Teitelboim gravity [28, 29] one can consider the path integral over two-dimensional metrics of fixed constant curvature with a boundary of fixed length [30, 31]. Our focus here is on pure three-dimensional gravity with negative cosmological constant. We will not attempt to compute the full path integral over all bulk geometries; instead we take spacetime to have fixed topology with a boundary on which the metric is that of a cylinder, $ds^2_{\partial M} = \frac{1}{\rho_c}(d\phi^2 + dt^2)$,[1] with circumference set by the variable parameter $\rho_c$. Our aim is to compute the classical algebra of observables, to define a Hilbert space that furnishes a unitary representation of this algebra, and to calculate the energy spectrum of the quantum theory in this sector. Attempting to define the full quantum theory is a much more difficult problem, which remains ill-understood even in the simpler case of an asymptotically AdS$_3$ boundary; see [32] for the current state of the art.

A major motivation for considering this problem is its connection to $T\overline{T}$-deformed CFTs [33–35]. We recall that this corresponds to a one parameter family of two-dimensional QFTs labelled by $\lambda$ whose action obeys the flow equation $\partial_\lambda S_\lambda = -\frac{1}{4}\int d^2x \det T$, where $T_{ij}$ is the stress tensor of the deformed theory with parameter $\lambda$.[2] Since $\det T$ is an irrelevant deformation of the $\lambda = 0$ seed theory, we expect dramatic effects in the UV. While these remain to be properly understood, the special properties of the $\det T$ operator allow certain quantities to be computed in the absence of this knowledge [33–40]. Most relevant for present purposes is the energy spectrum of the deformed theory on a spatial circle of circumference $2\pi$, for which there exists a general formula in terms of the spectrum of the seed theory. Focusing on CFTs and on the Virasoro descendants of the vacuum state, the energy and momentum eigenvalues of the seed CFT are

$$E = -\frac{c}{12} + N + \overline{N}, \quad P = N - \overline{N}, \tag{1.1}$$

where $(N, \overline{N})$ are the level numbers, which are non-negative integers. The $\lambda$-deformed energy spectrum is [33, 34]

$$\begin{aligned} E(\rho_c) &= \frac{c}{6\rho_c}\left(1 - \sqrt{1 + \rho_c - \frac{12}{c}\rho_c(N + \overline{N}) + \frac{36}{c^2}\rho_c^2(N - \overline{N})^2}\right) \\ &= -\frac{c}{6(1+\alpha)} + \frac{N + \overline{N}}{\alpha} + \frac{3\rho_c}{c}\frac{(N + \overline{N})^2 - \alpha^2(N - \overline{N})^2}{\alpha^3} + \mathcal{O}(c^{-2}). \end{aligned} \tag{1.2}$$

Here, we are expressing the parameter $\lambda$ in terms of bulk gravity language as

$$\lambda = \frac{4G\rho_c}{\pi}, \tag{1.3}$$

and we defined

$$\alpha = \sqrt{1 + \rho_c}. \tag{1.4}$$

Also, throughout this work $c$ is related to gravity variables by the standard Brown-Henneaux formula, $c = 3/2G$ in units where the AdS length is 1. In the second line of (1.2) we have expanded in $1/c$ while holding $\rho_c$ fixed. The momentum spectrum is unchanged, $P(\rho_c) = N - \overline{N}$, as follows from its integer quantization. For positive $\lambda$ the spectrum (1.2) exhibits unusual features due to the square root; energies can acquire imaginary parts and the spectrum is unbounded from below. Whether this signals incurable difficulties or not remains to be seen.

---

[1]We work in Euclidean signature, but given the product structure of our manifold the Wick rotation to Lorentzian signature is straightforward.

[2]Note that we often follow convention and refer to a "$T\overline{T}$" operator, even though we really mean $\det T$.

According to the proposal of [41], a $T\overline{T}$ deformed CFT is related to the holographically dual bulk theory with a radial cutoff; i.e. the bulk has a cylinder boundary as described above. The discrete quantum spectrum (1.2) is a prediction which remains to be verified directly in this bulk spacetime.[3] In the undeformed case with $\rho_c = 0$ the bulk spectrum corresponding to (1.2) is well understood: there are no local degrees of freedom but, as originally identified by Brown and Henneaux [42], there are boundary gravitons. Thus, we can think of our task as understanding these boundary gravitons, but now at a finite boundary. Similarly, in the asymptotically AdS case Brown and Henneaux demonstrated the emergence of the Virasoro algebras present in the CFT; how are these algebras deformed if we impose Dirichlet boundary conditions at finite $\rho_c$?[4]

Another goal is to gain a better understanding of observables that probe the UV structure of $T\overline{T}$-deformed theories; indeed there are reasons to expect (e.g. [46–48]) that the theory becomes nonlocal at a distance scale $\sqrt{\lambda}$. One route to gaining insight is by computing stress tensor correlation functions [49–52], or equivalently correlation functions involving boundary gravitons.

We also note that there is another way of thinking about the bulk description of a $T\overline{T}$ deformed CFT. Instead of working with a Dirichlet boundary condition on a cutoff surface one can impose a mixed boundary condition at the asymptotic AdS$_3$ boundary, which encodes the effect of adding the double trace $T\overline{T}$ interaction [43]. At the level of classical pure gravity these two descriptions are equivalent, as adding the $T\overline{T}$ deformation can be shown to coincide with subtracting the bulk action associated with the spacetime region between the cutoff and the asymptotic boundary [53]. In the present work we focus on the Dirichlet cutoff picture.

**Summary and results**

We now explain our approach and summarize our main results. We work in the general framework of the covariant phase approach to canonical quantization [54, 55], which is standard for this type of problem since it has the advantage of maintaining covariance. Covariant phase space in the presence of boundaries has been considered before (e.g. [22, 23, 56–58]) with a useful overview presented in [27]. However, we will work from the ground up, since our problem violates some of the assumptions that are typically made, for instance in [27].[5]

As stated, we consider the space of metrics with fixed topology while demanding that the metric on the boundary is $ds^2 = \frac{1}{\rho_c}(d\phi^2 + dt^2)$. Before initiating quantization, we first compute the Poisson bracket algebra of observables by putting the classical theory in canonical form. This involves characterizing the general classical solution (these are essentially the cutoff versions of the Bañados geometries [59]) and writing down a candidate symplectic form on this space. To render this form non-degenerate, we need to identifying appropriate gauge orbits, each of which defines a point in covariant phase space. The next step is to identify the spacetime vector fields $\xi$ that preserve the boundary metric. The associated charges $Q[\xi]$ constitute the set of classical observables of the theory. Unsurprisingly, they are given by integrals of the boundary (Brown-York [60]) stress tensor,

$$Q[\xi] = \frac{i}{2\pi} \int_0^{2\pi} T_{ti}\xi^i d\phi. \tag{1.5}$$

---

[3]At the classical level, the quasi-local energy of BTZ black holes was found in [41] to agree with 1.2, which in fact was one of the main arguments in favor of their proposal.

[4]Related questions were addressed in [43–45]; we discuss the relation of these references to the present work in the main text.

[5]Namely, because we are interested in the algebra of all boundary observables and not just symmetry generators, we consider diffeomorphism which move the location of the boundary and are not symmetries of the action.

Unlike the situation in asymptotically AdS, e.g. [61–63], or that in [27], these charges in general are not conserved in time and do not correspond to symmetries in any useful sense (as far as we can tell), with the exception of the energy and angular momentum charges. In particular, the diffeomorphism associated to a generic $\xi$ moves the location of the boundary and so does not leave the action invariant; indeed, if it did leave the action invariant Noether's theorem would yield a corresponding conserved charge. Instead, we simply think of the charges as being useful functions on phase space.

Following a standard line of logic that we review in the main text, the Poisson brackets of the charges may be extracted by computing the variation of the charges under boundary preserving diffeomorphisms according to the formula $i\{Q[\xi_1], Q[\xi_2]\} = -\delta_{\xi_1} Q[\xi_2]$. To justify this formula and work with it we need to appreciate an important subtlety, which is that the vector fields $\xi$ are "state dependent". This is to say that the space of $\xi$ fields depends on the particular solution to which the diffeomorphisms are being applied. The most important effect of this for us is that the Poisson bracket algebra is not a (centrally extended) Lie algebra; rather, the Poisson bracket of two charges results in an expression nonlinear — indeed nonpolynomial — in the charges.[6] The charge algebra, which is one of the main results of this work, is given in formulas (5.29).

We have not specified the spacetime topology to arrive at this result. If we take it to be Disk $\times \mathbb{R}$, we can expand the general result in $1/c$ around global AdS$_3$, which gives

$$i\{L_m, L_n\} = \frac{c}{12\alpha}(m^3 - m)\delta_{m+n} + \frac{m-n}{4\alpha^2}\left[(4 + 3\rho_c)L_{m+n} - \rho_c(2mn + 1)\bar{L}_{-m-n}\right] + \mathcal{O}(1/c),$$

$$i\{\bar{L}_m, \bar{L}_n\} = \frac{c}{12\alpha}(m^3 - m)\delta_{m+n} + \frac{m-n}{4\alpha^2}\left[(4 + 3\rho_c)\bar{L}_{m+n} - \rho_c(2mn + 1)L_{-m-n}\right] + \mathcal{O}(1/c),$$

$$i\{L_m, \bar{L}_n\} = -\frac{\rho_c}{4\alpha^2}\left[(m - n - 2mn^2)L_{m-n} + (m - n + 2m^2n)\bar{L}_{n-m}\right] + \mathcal{O}(1/c), \tag{1.6}$$

where the higher order contributions are nonlinear in the generators. Setting $\rho_c = 0$ (i.e. $\alpha = 1$) we recover the standard result for asymptotically AdS$_3$ gravity, namely a pair of Virasoro algebras with the Brown-Henneaux central charge $c = 3/2G$.

The second part of this work involves tackling the quantization problem in a systematic manner. Here we take inspiration[7] from the coadjoint orbit approach, which has been worked out in the asymptotically AdS$_3$ case, reproducing the results of Alekseev and Shatashvili, and of Witten [68, 69]. We note in particular [26], which used the Chern-Simons formulation to achieve this and worked out various implications (see also [70, 71], which adapted some of this discussion to $T\bar{T}$ deformed theories.) Here we work purely in the metric formulation, but the basic logic is the same. From our perspective, the point is that the symplectic form is highly nonlocal when expressed in terms of the Fourier modes of the boundary stress tensor, even though they are the physically relevant functions on space. This is a reflection of the fact that the natural symplectic manifold is not the space of all stress tensors but rather a single orbit, which is the same as the space of stress tensors related to each other by large gauge transformations (i.e. coordinate transformations that act nontrivially on the boundary). One may therefore take the gauge transformation parameters as coordinates on phase space.

To set this up, we first give a simple derivation of the Alekseev-Shatashvili symplectic form in a way that is straightforward to adapt to the case of a finite cutoff. As expected based on the fact that the charge algebra is non-polynomial in the presence of a radial cutoff, the

---

[6]The same phenomenon occurs, for the same reason, in AdS$_3$ higher spin gravity, where one encounters nonlinear $W$-algebras [64, 65]. Field dependence also enters in the central charge of BMS algebra in asymptotically flat space [66], which can be written as a Lie-algebroid.

[7]The usual starting point for coadjoint orbit quantization is a Lie group, such as the Virasoro group (see [67] which emphasizes the group theoretical point of view). Here we instead start from a space of metrics related by coordinate transformations, which has no natural Lie group structure because the space of coordinate transformations that preserve the boundary conditions depends on the metric on which they act.

deformed version of the Alekseev-Shatashvili symplectic form is significantly more complicated to obtain, and we do not yet have a closed form expression for it. We content ourselves with working out the first few orders in the large $c$ expansion. Within this perturbative procedure, we can proceed to quantize the theory by identifying a vacuum state as well as creation and annihilation operators for the left and right movers. We thus construct a Hilbert space and find an operator expression for the stress tensor.

In particular, we work out the leading interaction of the boundary gravitons, which is cubic and appears at order $1/\sqrt{c}$. Naively, this would seem to lead to $\mathcal{O}(1/\sqrt{c})$ terms in the energy spectrum; however we show that this contribution vanishes, in accord with the predicted $T\overline{T}$ spectrum (1.2). In fact, we find a unitary transformation on the Hamiltonian that eliminates any contribution of order $1/\sqrt{c}$ in the resulting operator.

We have not gone beyond this order in the $1/c$ expansion, as would be required to derive interesting new results, say for stress tensor correlators. We expect that this is possible once a suitable field redefinition is identified to simplify formulas; indeed we note that in the pure Virasoro case a field redefinition renders the symplectic form and stress tensor purely quadratic, which (as noted in [26]) makes it easy to compute things like the partition function. The fact that the $T\overline{T}$ spectrum (1.2) is known strongly suggests that a judicious field redefinition will yield major simplifications here as well.

We leave the search for such a field redefinition to future work, but close the main text with a discussion of how we expect things to work at higher orders. Essentially, we find that the $1/c$ expansion of the Poisson algebra, along with some plausible assumptions about the operators in the theory, implies the $\mathcal{O}(1/c)$ correction to the energy expected from (1.2). Furthermore, we observe that the $T\overline{T}$ spectrum (1.2) predicts the existence of an operator unitarily equivalent to $-\frac{c}{12} + \hat{N} + \hat{\overline{N}}$ where $\hat{N}$ and $\hat{\overline{N}}$ are level number operators. This statement is similar to [40] where the $T\overline{T}$ deformation was found equivalent, in part, to a unitary transformation. The existence of such a unitary operator is supported to low order by our direct perturbative calculations.

**Outline**

The rest of this paper is organized as follows. In Section 2 we review the canonical formulation of classical mechanics and field theory using the covariant phase space method. We set up the classical formulation of GR with a Dirichlet boundary in Section 3. We discuss the gravitational symplectic form and find that large diffeomorphisms are generated by charges that can be written in terms of the stress tensor on the spatial boundary. These concepts are illustrated in Section 4 for an asymptotically AdS$_3$ spacetime. The explicit expressions for the charges and their Virasoro algebra are reviewed, as well as the Alekseev-Shatashvili symplectic form in terms of the gauge transformations that parameterize the coadjoint orbit. We also set up the perturbative expansion that we will use for the AdS$_3$ spacetimes with a finite cylinder boundary. Such spacetimes are reviewed in Section 5. After writing the geometry, boundary charges, and the spacetime vectors that preserve the boundary metric, we calculate how these transformations act on the charges. This leads to the nonlinear Poisson bracket algebra in equation (5.29), which is the central result of the first part of this work. Expanding in orders of $1/c$ around the global AdS values, we obtain the deformed algebra of "Virasoro" generators $L_m$ and $\overline{L}_m$. In Section 6 we prepare to quantize this theory perturbatively around AdS$_3$ with a cutoff. We find the Hamiltonian, the momentum and the symplectic form up to third order. The expressions are simplified drastically using a field redefinition to Darboux coordinates, after which all is in place for quantization in Section 7. We find the symplectic form and boundary charges in terms of creation and annihilation operators. In particular, we calculate the Hamiltonian up to third order, allowing us to find the spectrum at order $1/\sqrt{c}$ in agreement with the

known formula (1.2). In Section 8 we discuss some observations which hint towards a possible all-orders calculation of the spectrum and Section 9 contains our concluding discussion.

## 2 Canonical formulation and covariant phase space

### 2.1 Canonical formulation

We begin by recalling some elements of the symplectic formulation of classical dynamics that will be needed for what follows. We start with a symplectic manifold $\Gamma$, which by definition is an even dimensional manifold equipped with a closed, non-degenerate two-form $\Omega$. For the immediate discussion we take $\Gamma$ to be finite dimensional, $\dim(\Gamma) = 2n$, anticipating the extension to the infinite dimensional case relevant to our field theory context. Closure is defined as $\delta\Omega = 0$, where $\delta$ is the exterior derivative on $\Gamma$.[8] Non-degeneracy means that the equation $i_V\Omega = 0$ implies $V = 0$, where $V$ is a vector field on $\Gamma$ and $i_V$ is the standard contraction operation taking a $p$-form to a $(p-1)$-form, given in coordinates momentarily.

We now define local coordinates $\{q^i\}$, $i = 1, 2, \ldots 2n$, in terms of which $\Omega = \frac{1}{2}\Omega_{ij}\delta q^i \wedge \delta q^j$ with $\Omega_{ji} = -\Omega_{ij}$. The contraction operation is $(i_V\Omega)_j = V^i\Omega_{ij}$. We define $\Omega^{ij}$ via $\Omega^{ik}\Omega_{kj} = \delta^i_j$, noting that $\Omega_{ij}$ is invertible by the non-degeneracy assumption. $\Omega^{ij}$ is used to define the Poisson bracket. Namely, given two functions on phase space $F(q)$ and $G(q)$ we define

$$\{F, G\} = \Omega^{ij}\partial_i F\partial_j G, \tag{2.1}$$

where $\partial_i = \frac{\partial}{\partial q^i}$.

Given a vector field $V = V^i\partial_i$ we have the associated Lie derivative $\mathcal{L}_V$. We recall that it obeys

$$\mathcal{L}_V\mathcal{L}_W - \mathcal{L}_W\mathcal{L}_V = \mathcal{L}_{[V,W]}, \tag{2.2}$$

where $[V, W]^i = V^j\partial_j W^i - W^j\partial_j V^i$ is the commutator. When acting on differential forms it is extremely useful to work with the Cartan formula

$$\mathcal{L}_V X = (\delta i_V + i_V\delta)X, \tag{2.3}$$

where $X$ denotes an arbitrary differential form over $\Gamma$.

Infinitesimal canonical transformations correspond to flows generated by vector fields $V$ that preserve the symplectic form in the sense that $\mathcal{L}_V\Omega = 0$. To obtain such vector fields, let $F$ be a function on phase space and define its associated "Hamiltonian vector field" $V_F$ as

$$V_F^i = \Omega^{ij}\partial_j F. \tag{2.4}$$

This expression may be inverted as

$$\delta F = -i_{V_F}\Omega. \tag{2.5}$$

Using this, along with the Cartan formula and the closure of $\Omega$, we have

$$\mathcal{L}_{V_F}\Omega = \delta i_{V_F}\Omega = -\delta^2 F = 0. \tag{2.6}$$

The Poisson bracket may now be written in various forms as

$$\begin{aligned} \{F, G\} &= i_{V_G}\delta F = -i_{V_F}\delta G \\ &= \mathcal{L}_{V_G}F = -\mathcal{L}_{V_F}G. \end{aligned} \tag{2.7}$$

---

[8]We reserve the symbol $d$ to denote the exterior derivative on spacetime.

Using these formulas and (2.2) it is straightforward to verify that the Poisson bracket obeys the Jacobi identity.

As an aside, we note Darboux's theorem, which is the statement that we can choose local coordinates $(P^a, Q^a)$, $a = 1, 2, \ldots n$, such that $\Omega = \delta P^a \wedge \delta Q^a$ which produces the standard Poisson brackets.

In the canonical formulation of a classical system the equations of motion take the form

$$\dot{q}^i = \{q^i, H\} = \Omega^{ij} \partial_j H, \tag{2.8}$$

where $H = H(q)$ is by definition the Hamiltonian. A function $Q = Q(q, t)$, where a possible explicit dependence on time is indicated, is conserved if

$$\frac{dQ}{dt} = \frac{\partial Q}{\partial t} + \{Q, H\} = 0. \tag{2.9}$$

We now explain how we compute charge algebras in the context of asymptotic symmetries. The definition (2.1) is inconvenient due to the need to invert $\Omega_{ij}$. It is more convenient to use (2.7). We also adopt the notation $\delta_{V_F} G = \mathcal{L}_{V_F} G$, so that the Poisson bracket becomes

$$\{F, G\} = \delta_{V_G} F = -\delta_{V_F} G. \tag{2.10}$$

In the context of asymptotic symmetries, we start by identifying spacetime diffeomorphisms that preserve some stated boundary conditions. These give rise to vector fields on phase space $V^a$ ($a$ labels the vector field, not a component), which act as canonical transformations. We then deduce that $V^a$ are Hamiltonian vector fields, in the sense of (2.4). The corresponding function $F$ on phase space is called the associated charge $Q^a$. Given the explicit form of $V^a$ and $Q^a$ we can compute $\delta_{V^a} Q^b$, and thereby deduce the Poisson bracket

$$\{Q^a, Q^b\} = -\delta_{V^a} Q^b. \tag{2.11}$$

Suppose that the vector fields $V^a$ obey an algebra

$$[\mathcal{L}_{V^a}, \mathcal{L}_{V^b}] = if^{abc} \mathcal{L}_{V^c}, \tag{2.12}$$

where the structure constants $f^{abc}$ are possibly nontrivial functions on phase space. We can use this to infer the Poisson bracket algebra of the charges $Q^a$, up to central terms. To this end, let $F$ be an arbitrary function on phase space and use the Jacobi identity to write

$$\{\{Q^a, Q^b\}, F\} = \{\{Q^a, F\}, Q^b\} - \{\{Q^b, F\}, Q^a\}. \tag{2.13}$$

Using (2.7) this may be written as

$$\{\{Q^a, Q^b\}, F\} = -[\mathcal{L}_{V^a}, \mathcal{L}_{V^b}] F = -if^{abc} \mathcal{L}_{V^c} F. \tag{2.14}$$

Recalling $\{Q^c, F\} = -\mathcal{L}_{V^c} F$ we deduce that we must have

$$\{Q^a, Q^b\} = if^{abc} Q^c + Z^{ab}, \tag{2.15}$$

where $Z^{ab}$ is a central term that has vanishing Poisson brackets with everything. In general, this is not an ordinary (centrally extended) Lie algebra, since the structure constants $f^{abc}$ can be field dependent, i.e. be nontrivial functions on phase space.

The usual context in which one does obtain an ordinary Lie algebra (with possible central extension) is as follows. As a concrete and relevant example [42], it is useful to have in mind the theory of gravity coupled to matter in an asymptotically AdS spacetime. In that case, and more generally, we have some asymptotic boundary conditions which are preserved

by diffeomorphisms generated by some fixed vector fields $\xi^a$. By "fixed" we mean that the same vector fields may be used for all solutions in the theory that respect whatever boundary conditions have been imposed. For example, in the asymptotically AdS$_{d+1}$ case with $d > 2$, the vector fields can be chosen to be Killing vectors of global AdS; acting on a general solution these vector fields do not leave the solution invariant, but they do preserve the relevant boundary conditions.[9] These spacetime vector fields will obey a Lie algebra $[\xi^a, \xi^b] = if^{abc}\xi^c$ with fixed structure constants $f^{abc}$. Furthermore, when acting on covariant objects (built out of the metric and curvature tensor and covariant derivatives of matter fields), the phase space vector fields corresponding to the $\xi^a$ will obey the same Lie algebra as the $\xi^a$, and from this it follows that so too will $Q^a$, up to a possible central extension. So under these conditions, we know that the Poisson bracket algebra of the charges will coincide with the Lie algebra of the spacetime vector fields up to a possible central extension, and so all that remains is to compute the central extension.

The argument of the previous paragraph does not go through if the vector fields $\xi^a$ are field dependent, or more precisely if the structure constants in $[\xi^a, \xi^b] = if^{abc}\xi^c$ are non-constant on phase space. The field dependence shows up in the fact that $Q^a$ will now obey a nonlinear algebra. This nonlinear algebra may be computed from (2.11).

## 2.2 Covariant phase space

We adopt the widely used method of covariant phase space, since it allows for a canonical formalism without sacrificing manifest spacetime symmetry. Essentially all that we will need is contained in the elegant original discussion in [54]. The basic idea is to think of phase space as the space of classical solutions (modulo gauge transformations). The usual $p$'s and $q$'s are thought of as particular coordinates on the phase space, corresponding to initial data on some chosen Cauchy slice, but to retain manifest symmetry one can refrain from committing to such coordinates or to a particular Cauchy slice.

We now collect the main formulas and points of notation. We denote the collection of dynamical fields as $\phi^a(x)$, which are subject to some classical equations of motion. Temporarily ignoring issues of gauge redundancy, we define phase space as the space of classical solutions obeying specified boundary conditions. If $\delta_\xi \phi^a$ represents some variation of fields on this space (that is, $\delta_\xi \phi^a$ is a solution of the equations of motion linearized around a particular solution) we define the corresponding vector field

$$V_\xi = \int_M dx\, \delta_\xi \phi^a(x) \frac{\delta}{\delta \phi^a(x)}, \qquad (2.16)$$

where the integral is over all of spacetime $M$. As usual, we define a dual space of differential forms and an exterior derivative $\delta$, so that $\delta\phi^a(x)$ is a one-form. We write $i_V$ to denote contraction with respect to the vector field $V$ as in the previous section. For example, the contraction of the vector field $V_\xi$ with the one-form $\delta\phi^a(x)$ is

$$i_{V_\xi} \delta\phi^a(x) = \mathcal{L}_V \phi^a(x) \equiv \delta_\xi \phi^a(x), \qquad (2.17)$$

which is the notation introduced in the last section extended to the covariant phase space. To clarify notation, we emphasize that if $\Psi$ is a $p$-form on phase space, then $\delta\Psi$ is a $(p+1)$-form while $\delta_\xi \Psi$ is a $p$-form; that is, $\delta_\xi \Psi$ represents a particular variation and not an exterior derivative.

It is important to keep straight the distinction between operations in spacetime versus those on phase space. In spacetime we have vector fields $\xi = \xi^\mu \frac{\partial}{\partial x^\mu}$, differential $p$-forms

---

[9]For AdS$_3$ there is an enhancement due to the inclusion of vector fields that act as conformal Killing vectors of the boundary metric, but again these can be taken to be fixed vector fields.

$\Phi = \frac{1}{p!}\Phi_{\mu_1\ldots\mu_p}dx^{\mu_1}\wedge\ldots\wedge dx^{\mu_p}$, the contraction operation $i_\xi$, and the exterior derivative $d$. Associated to the vector field $\xi$ is the Lie derivative $\mathcal{L}_\xi$. Acting on a spacetime differential form $\Phi$ the Cartan formula is

$$\mathcal{L}_\xi \Phi = (di_\xi + i_\xi d)\Phi.\qquad(2.18)$$

On phase space we have vector fields $V_\xi$ as in (2.16), differential $p$-forms

$$\Psi = \frac{1}{p!}\int_{M^p} dx_1\ldots dx_p \Psi_{a_1\ldots a_p}(x_1,\ldots,x_p)\delta\phi^{a_1}(x_1)\wedge\ldots\wedge\delta\phi^{a_p}(x_p),\qquad(2.19)$$

the contraction operation $i_{V_\xi}$, and the exterior derivative $\delta$. Acting on a phase space differential form $\Psi$ the Cartan formula is

$$\mathcal{L}_V \Psi = (\delta i_V + i_V \delta)\Psi.\qquad(2.20)$$

We will adopt the convention that $d$ and $\delta$ commute: $d\delta = \delta d$.

An important class of field variations corresponds to an infinitesimal coordinate transformation $x^\mu \to x^\mu + \xi^\mu(x)$,

$$\delta_\xi \phi^a(x) = \mathcal{L}_\xi \phi^a(x).\qquad(2.21)$$

We write $V_\xi$ as the corresponding vector field on phase space, as defined in (2.16). For spacetime vector fields $\xi$ that are field independent in the sense of not varying over phase space we have the useful equality

$$\mathcal{L}_\xi \Phi = \mathcal{L}_{V_\xi}\Phi,\qquad(2.22)$$

where $\Phi$ is a "covariant tensor" (e.g. a local expression built out the metric and covariant derivatives of fields). However, as mentioned in the previous section, we will be working with field dependent vector fields $\xi$ that do vary over phase space, and it is important to note that (2.22) does not hold in such cases. The issue can be appreciated from (2.20): $\delta$ in the first term acts nontrivially on the $\xi$ in $V_\xi$, which spoils the equality with $\mathcal{L}_\xi$. Note that this term is absent if $\Phi$ is a 0-form on phase space, in which case the relation (2.22) does hold.

## 3 Gravity with Dirichlet boundary conditions

In this section we discuss some general issues regarding gravity with a Dirichlet boundary condition on the metric. From here on we work in Euclidean signature, but since we work with spacetimes $M$ with a single connected boundary of the form $\partial M = \partial\Sigma \times \mathbb{R}$ the Wick rotation to Lorentzian is obvious. Our choice to work in Euclidean signature is essentially just a notational choice, made for easy comparison with standard CFT formulas.

In spacetime dimension $D = d + 1 > 3$ imposing Dirichlet boundary conditions on some cutoff surface in Euclidean signature gravity is incompatible with a sensible perturbation theory around a given background solution; see [21] for a review. The case of $D = 3$ is of course special given the absence of local degrees of freedom, and indeed we will explicitly implement a sensible perturbation theory in this work. In Lorentzian signature one sometimes encounters superluminal propagation with respect to the cutoff boundary metric [20], a feature that is relevant [41] to the interpretation of $T\overline{T}$ deformed theories as being nonlocal.

### 3.1 Gravity with a boundary

We consider the spacetime manifolds $M$ which have boundary topology[10] $\partial M = \partial\Sigma \times \mathbb{R}$, where $\Sigma$ should be thought of as a slice of $M$, restricted only to have boundary $\partial\Sigma$ which is connected and compact. We choose a radial coordinate $\rho$ such that $\partial M = \partial\Sigma \times \mathbb{R}$ lies at $\rho = \rho_c$. We take $x^i$ to denote coordinates on $\partial M$. In a vicinity of the boundary it will be convenient to adopt Gaussian normal coordinates such that $g_{\rho i} = 0$. More precisely, in a vicinity of the boundary we take

$$ds^2 = \frac{\ell^2 d\rho^2}{4\rho^2} + g_{ij}(\rho, x)dx^i dx^j, \tag{3.1}$$

where we take the region interior to the boundary to be $\rho > \rho_c$. The metric on $\partial M$ is therefore written as $g_{ij} = g_{ij}(\rho_c, x^k)$. The choice $g_{\rho\rho} = \frac{\ell^2}{4\rho^2}$ is convenient because in the asymptotically AdS case in which $\rho_c \to 0$ one has a small $\rho$ expansion[11] $\rho g_{ij} \sim \rho^0 + \rho + \dots$.

We consider the Einstein-Hilbert action with cosmological constant $\Lambda$ in Euclidean signature[12]

$$S = -\frac{1}{16\pi G} \int d^{d+1}x \sqrt{g}\left(R - 2\Lambda\right) + S_{\text{bndy}}. \tag{3.2}$$

Our main example will involve negative cosmological constant, in which case we write $\Lambda = -d(d-1)/2\ell^2$. Einstein's equations are then

$$R_{\mu\nu} - \frac{1}{2}Rg_{\mu\nu} = \frac{d(d-1)}{2\ell^2}g_{\mu\nu}. \tag{3.3}$$

We henceforth set $\ell = 1$.

We fix the metric on the boundary by imposing the boundary condition $\delta g_{ij}\big|_{\rho_c} = 0$. Given our coordinate choice (3.1) this actually imposes $\delta g_{\mu\nu}\big|_{\rho_c} = 0$. Stationarity of the action requires us to include the Gibbons-Hawking terms

$$S_{\text{bndy}} = -\frac{1}{8\pi G} \int_{\partial M} d^d x \sqrt{g^{(d)}} K + S_{\text{ct}}, \tag{3.4}$$

where we also allow for additional counterterms $S_{\text{ct}}$, and $\sqrt{g^{(d)}} = \sqrt{\det g_{ij}}$. In the coordinates (3.1) the extrinsic curvature is

$$K_{ij} = -\rho \partial_\rho g_{ij}, \tag{3.5}$$

and $K = g^{ij}K_{ij}$.

The boundary stress tensor is defined in terms of the on-shell variation of the action [60]

$$\delta S = \frac{1}{4\pi} \int_{\partial M} d^d x \sqrt{g^{(d)}} T^{ij} \delta g_{ij}. \tag{3.6}$$

---

[10]Since essentially all our work will be localized to the boundary of spacetime, it is sufficient for our purposes to specify only the boundary topology.

[11]In $d = 2$ and pure gravity the series contains only 3 terms. For $d > 2$ there are more terms, as well as possibly $\log\rho$ terms.

[12]Here $g$ in $\sqrt{g}$ refers to the full $d+1$ dimensional metric. When we want to refer to the $d$-dimensional metric on a fixed $\rho$ surface we will always make this explicit by writing $g_{ij}$ and $\sqrt{g^{(d)}}$.

The boundary stress tensor is covariantly conserved with respect to the boundary metric, $\nabla_i T^{ij} = 0$. From this it follow that if $\xi^i$ is a Killing vector of the boundary metric, $\delta_\xi g_{ij} = \nabla_i \xi_j + \nabla_j \xi_i = 0$, then the corresponding charge[13]

$$Q[\xi] = \frac{i}{2\pi} \int_{\partial\Sigma} d^{d-1}x \sqrt{g^{(d-1)}} T_{ij} n^i \xi^j \tag{3.7}$$

is conserved under time evolution. Here we have taken $\sqrt{g^{(d-1)}}$ to be the volume element on $\partial\Sigma$, and $n^i$ is the unit vector in $\partial M$ normal to $\partial\Sigma$. For example, if we take the boundary $\partial M$ to be $S^{d-1} \times \mathbb{R}$ with line element $ds^2_{\partial M} = dt^2 + d\Omega^2_{d-1}$ then we obtain a conserved energy $E$ corresponding to time translations and conserved angular momenta $J^{ab}$ corresponding to SO(d) rotations.

We will be interested in boundary condition preserving diffeomorphisms, $\delta_\xi g_{\mu\nu} = \nabla_\mu \xi_\nu + \nabla_\nu \xi_\mu$ such that $\delta_\xi g_{\mu\nu}\big|_{\rho_c} = 0$. Vector fields $\xi^\mu$ that achieve this have to be field dependent in general, which is to say that in order to respect the boundary conditions, $\xi^\mu$ must change when we change the solution under consideration. Another salient remark is that we are taking the boundary to lie at fixed *coordinate* location $\rho = \rho_c$. A diffeomorphism such that $\xi^\rho\big|_{\rho_c} \neq 0$ may then be interpreted as moving the physical location of the boundary. In general, these are not to be thought of as gauge transformations or symmetries, but rather as particular transformations in phase space.

Given a general background solution it is not easy to find all diffeomorphisms that preserve the boundary conditions. But since the problem involves respecting the boundary conditions we can work locally near the boundary. Our strategy will be to define an initial time surface on the boundary, choose arbitrary functions $\xi^i$ on that surface, and then fix $\xi^\mu$ everywhere in the vicinity of the boundary by demanding $\delta_\xi g_{\mu\nu}\big|_{\rho_c} = 0$. This procedure will be made fully explicit in our main case of interest, namely AdS$_3$ gravity.

## 3.2 Symplectic form and boundary charges

To define the symplectic form it is convenient to think of the Lagrangian density $L$ as a differential $(d+1)$-form on spacetime, writing

$$S = \int_M L + \int_{\partial M} L_b, \tag{3.8}$$

where we also included a boundary term. The variation of the Lagrangian density can always be written in the form

$$\delta L = E_a \delta\phi^a + d\Theta. \tag{3.9}$$

This holds for a general theory and we are denoting the collection of all dynamical fields as $\phi^a$. See [72] for a pedagogical discussion and further references. Also, in this formula — but nowhere else in the text unless explicitly indicated otherwise — the symbol $\delta$ denotes a general off-shell variation of the configuration space fields rather than a phase space exterior derivative. The Euler-Lagrange equations are by definition $E_a = 0$. Of course, (3.9) only defines $\Theta$ up to the addition of a closed form. To fix this ambiguity we require that $\Theta$ be a local covariant expression, and that the on-shell variation of the action, $\delta S = \int (\Theta + \delta L_b)$, vanishes under variations that preserve the boundary conditions. We refer to [27] for more discussion, and just note that here we will use an explicit formula that fulfills all requirements.

Going forward we work in the framework of covariant phase space, so that $\delta\phi^a$ is a 1-form defined on the space of classical solutions. Since $\Theta$ is linear in field variations it is also a 1-form

---

[13]The factor of $i$ is due to our choice of Euclidean signature.

on this space. On the other hand, $\Theta$ is a $d$-form on spacetime. To define the (pre)symplectic form $\Omega$ we integrate the exterior derivative of $\Theta$ over a Cauchy surface $\Sigma$,[14]

$$\Omega = i \int_{\Sigma} \delta\Theta \,. \tag{3.10}$$

By construction, $\Omega$ is a closed 2-form on the space of classical solutions. This $\Omega$ cannot yet be identified as a symplectic form since it is degenerate: diffeomorphism invariance of the action implies that $i_{V_\xi}\Omega = 0$ where $\xi$ is any vector field that vanishes at the boundary. We therefore define equivalence classes of classical solutions so as to remove the degenerate directions. In the context of the present work, we emphasize that solutions related by "large" gauge transformations, which here refers to coordinate transformations that do not vanish on the boundary, need not lie in the same equivalence class because the vector fields that take us among such solutions are not degenerate with respect to the symplectic form. This is the mechanism by which "pure gauge" degrees of freedom become physical in the presence of a boundary. Anticipating that we will later project out these pure-gauge directions, we will be loose with terminology by referring to $\Omega$ as the symplectic form.

Reference [27] gives a clear and useful exposition of covariant phase space in theories with a boundary, but we should highlight some points that render some of their results inapplicable to the problem we wish to solve here. In [27] emphasis is placed on spacetime symmetries and their associated conserved charges. Given a covariant Lagrangian defined on a space with boundary, the action is only invariant under coordinate transformations that map the boundary to itself (i.e. interpreted in an active sense, they do not move the boundary). Also, the authors primarily restrict attention to diffeomorphism vector fields $\xi$ that do not vary on phase space. Here, our goal is not just to identify symmetries but rather to lay the groundwork for quantization, which involves working with arbitrary functions and transformations on phase space. For this reason, we will be dealing with vector fields that violate both of the conditions mentioned above, although vector fields corresponding to symmetries are present as special cases.

Coming back to (3.10), we can alternatively write $\Omega$ as

$$\Omega = i \int_{\Sigma} d\Sigma_\alpha \sqrt{g} J^\alpha \,, \tag{3.11}$$

where the symplectic current $J^\alpha$ is the Hodge dual of $\delta\Theta$,

$$\delta\Theta = J^\alpha \sqrt{g} (d^d x)_\alpha \,. \tag{3.12}$$

Here

$$(d^{d+1-p}x)_{\mu_1\dots\mu_p} = \frac{1}{p!(d+1-p)!} \varepsilon_{\mu_1\dots\mu_p \nu_{p+1}\dots\nu_{d+1}} dx^{\nu_{p+1}} \wedge \dots \wedge dx^{\nu_{d+1}} \,, \tag{3.13}$$

with $\varepsilon$ the fully antisymmetric symbol with entries $\pm 1$ and $0$ so $\sqrt{g}d^{d+1}x$ is the spacetime volume form. The symplectic current is conserved on-shell, $\nabla_\alpha J^\alpha = 0$. This follows from the identity $d\delta\Theta = \nabla_\alpha J^\alpha \sqrt{g} d^{d+1}x$, along with $d\delta\Theta = \delta d\Theta = \delta^2 L = 0$. So with suitable boundary conditions on $\partial M$ — Dirichlet in our case — $\Omega$ is the same for any choice of $\Sigma$.

The symplectic current $J^\alpha$ was computed in [54],

$$J^\alpha = \frac{1}{16\pi G} \left[ \delta\Gamma^\alpha_{\mu\nu} \wedge \left( \delta g^{\mu\nu} + \frac{1}{2} g^{\mu\nu} \delta\ln g \right) - \delta\Gamma^\nu_{\mu\nu} \wedge \left( \delta g^{\alpha\mu} + \frac{1}{2} g^{\alpha\mu} \delta\ln g \right) \right] \,. \tag{3.14}$$

---

[14] We have inserted a factor of $i$ due to the fact that we are working in Euclidean signature.

Let $\xi^\mu$ be a spacetime vector field corresponding to the infinitesimal coordinate transformation $x^\mu \to x^\mu + \xi^\mu$, under which the metric varies as

$$\delta_\xi g_{\mu\nu} = \nabla_\mu \xi_\nu + \nabla_\nu \xi_\mu. \tag{3.15}$$

We denote the corresponding vector field on phase space as $V_\xi$. A key relation is given by contracting this vector field with the symplectic current, $i_{V_\xi} J^\alpha$, and identifying it as the derivative of an antisymmetric tensor. The result is [54]

$$i_{V_\xi} J^\alpha = -\frac{1}{16\pi G} \nabla_\nu X^{\alpha\nu}, \tag{3.16}$$

with

$$X^{\alpha\nu} = \left[ (\nabla_\mu \delta g^{\mu\nu} + \nabla^\nu \delta \ln g) \xi^\alpha + \nabla^\nu \delta g^{\mu\alpha} \xi_\mu - \nabla_\mu \xi^\alpha \delta g^{\mu\nu} - \frac{1}{2} \nabla^\nu \xi^\alpha \delta \ln g \right] - (\alpha \leftrightarrow \nu). \tag{3.17}$$

For completeness, we provide a few more comments on the above in Appendix B. Defining the $(d-1)$-form

$$X = X^{\alpha\nu} \sqrt{g} (d^{d-1}x)_{\alpha\nu} \tag{3.18}$$

we have

$$i_{V_\xi} \Omega = \frac{i}{16\pi G} \int_\Sigma dX = \frac{i}{16\pi G} \int_{\partial\Sigma} X. \tag{3.19}$$

Here $\partial\Sigma$ lies at fixed $\rho$, and for simplicity we also take it to lie at fixed $t$. This gives

$$i_{V_\xi} \Omega = \frac{i}{16\pi G} \int_{\partial\Sigma} d^{d-1}x \sqrt{g} X^{\rho t}. \tag{3.20}$$

With the boundary condition $\delta g_{\mu\nu}|_{\rho_c} = 0$ it is simple to evaluate $X^{\rho t}$, since $\delta g_{\mu\nu}$ must appear with a $\rho$-derivative in order not to vanish. So in (3.17) we can make the replacements $\nabla_\mu = \delta_{\mu,\rho} \partial_\rho$ and $\nabla^\mu = 4\rho^2 \delta_{\mu,\rho} \partial_\rho$. We also note that $\partial_\rho \delta g_{\rho\mu}|_{\rho_c} = \partial_\rho \delta g^{\rho\mu}|_{\rho_c} = 0$. Almost all terms vanish, and we find

$$\begin{aligned} X^{\rho t} &= -\nabla^\rho \delta \ln g\, \xi^t - \nabla^\rho \delta g^{\mu t} \xi_\mu \\ &= -4\rho_c^2 g^{ij} \partial_\rho \delta g_{ij} \xi^t + 4\rho^2 \partial_\rho \delta g_{ij} g^{jt} \xi^i. \end{aligned} \tag{3.21}$$

It is convenient to separate the coordinates on the boundary into space and time as $x^i = (t, x^a)$. It is also convenient (though not necessary) to choose these coordinates so that $g_{ta}|_{\partial\Sigma} = 0$. Collecting the terms multiplying $\xi^t$ and $\xi^a$ we have

$$X^{\rho t} = -4\rho_c g^{tt} \left( \delta(K_{tt} - g_{tt} K) \xi^t + \delta K_{ta} \xi^a \right), \tag{3.22}$$

where the extrinsic curvature is given in (3.5). The variation of the boundary stress tensor is

$$\delta T_{ij} = \frac{1}{4G} \delta(K_{ij} - K g_{ij}). \tag{3.23}$$

Using this, along with $\sqrt{g} g^{tt} = \frac{1}{2\rho} \sqrt{g^{(d-1)}} n^t$, where $n^t = 1/\sqrt{g_{tt}}$ is the unit normal, we find

$$\sqrt{g} X^{\rho t} = -8G \sqrt{g^{(d-1)}} \delta T_{ti} n^t \xi^i. \tag{3.24}$$

We then obtain the main result of this section

$$i_{V_\xi} \Omega = -\delta Q[\xi], \tag{3.25}$$

with[15]

$$Q[\xi] = \frac{i}{2\pi} \int_{\partial \Sigma} d^{d-1}x \sqrt{g^{(d-1)}} T_{ij} n^i \xi^j. \tag{3.26}$$

Several important comments are in order. First, as usually defined the boundary stress tensor contains certain terms that depend solely on the boundary metric and not its radial derivatives. Such terms are typically included in order to obtain finite conserved charges when the boundary surface is taken to infinity [61], or from other considerations [73]. We are free to include such terms in $T_{ij}$ since they have no variation under our boundary conditions. In other words, our arguments determine the charges $Q[\xi]$ up to a contribution that is constant on phase space.

Second, the change in the metric (3.15) induced by $\xi^\mu$ must respect the boundary conditions $\delta g_{\mu\nu}|_{\rho_c} = 0$, and as we have discussed this requires $\xi^\mu$ to be field dependent; i.e. $\xi^\mu$ is non-constant on phase space. On the other hand, in passing from (3.24) to (3.25) we evidently took $\delta \xi^i|_{\rho_c} = 0$ in order to write the result as $\delta(\ldots)$. Why doesn't this contradict the statement that $\xi^\mu$ is non-constant on phase space? The point is that although $\xi^\mu$ indeed has to vary on phase space in order to preserve the boundary conditions, we are free to choose $\xi$ arbitrarily on $\partial \Sigma$ for a fixed slice, as we will see. The (field dependent) form of $\xi^i$ away from $\partial \Sigma$ is then determined by enforcing the boundary conditions which, as we will derive below, take the form of a differential equation for $\partial_t \xi^\mu$. However, there is no obstacle to taking $\delta \xi^i|_{\partial \Sigma} = 0$ on some fixed slice. We will make this explicit when we compute the boundary condition preserving coordinate transformations.

We emphasize that the charge $Q[\xi]$ is not conserved under time evolution unless $\xi^i$ is a Killing vector of the boundary metric. This follows from the fact that $\nabla^i(T_{ij}\xi^j) = \frac{1}{2} T_{ij} (\nabla^i \xi^j + \nabla^j \xi^i)$, where we used conservation of the boundary stress tensor. The relevance of the general $Q[\xi]$ is that it is the function on phase space whose corresponding Hamiltonian vector field is $V_\xi$, whose action matches that of $\xi$ for 0-forms over phase space, as was discussed at the end of Section 2.

We also note that in certain cases, such as AdS$_3$ with an asymptotic cylinder boundary, the boundary stress tensor is traceless, in which case there are additional conserved charges $Q[\xi]$, with $\xi^i$ a conformal Killing vector of the boundary metric. In the asymptotic AdS$_3$ case one thereby obtains the conserved charges that appear in the asymptotic Virasoro algebra. In the general case we will obtain an algebra of charges $Q[\xi]$, but we refrain from referring to it as a symmetry algebra since the charges are not conserved in any useful sense.[16]

## 3.3 Pure AdS$_3$ gravity and its connection to the $T\overline{T}$ deformation

One motivation for this work is the result of Zamolodchikov and Smirnov [33,34] on the energy spectrum of $T\overline{T}$ deformed CFT which, combined with the conjecture [41] gives a prediction for the spectrum of pure AdS$_3$ gravity with Dirichlet boundary conditions. We now review relevant aspects of this story, mostly following [41,43,49,53].

The $T\overline{T}$ deformation describes a one-parameter family of two-dimensional quantum field theories labelled by $\lambda$ whose action obeys the flow equation[17]

$$\frac{dS_\lambda}{d\lambda} = -\frac{1}{4} \int d^2x \sqrt{\gamma} \det T^i_j, \tag{3.27}$$

---

[15]The covariant form of the result makes it clear that it does not depend on our assumption $g_{ta}|_{\partial \Sigma} = 0$.

[16]Here we mean that while we could include an explicit time dependence in the definition of the charges in order to make them conserved, determining what this time dependence must be involves solving the equations of motion, which defeats the purpose.

[17]Our conventions follow [53].

where it is important to keep in mind that the stress tensor itself depends on $\lambda$. We restrict attention to the case where the theory at $\lambda = 0$ is a CFT, and also take the background metric $\gamma_{ij}$ to be flat. We are writing the metric as $\gamma_{ij}$ because it will be related by a rescaling to the bulk metric $g_{ij}$. For a deformed CFT, the parameter $\lambda$, which has mass dimension $-2$, is the only dimensionful scale, and so the statement of dimensional analysis $S_{\sigma^2\lambda}(\sigma^2 g_{\mu\nu}) = S_\lambda(g_{\mu\nu})$ together with the definition (3.6) of the stress tensor imply that (3.27) is equivalent to

$$T_i^i = \pi\lambda \det T_j^i, \tag{3.28}$$

up to total derivatives.

The simplest route [49] to seeing the connection with a bulk description is to note that (3.28) is equivalent to one of the Einstein equations for pure AdS$_3$ gravity with Dirichlet boundary conditions at a radial cutoff, as we now review.

Writing the metric as in (3.1), the Einstein equations $R_{\mu\nu} = -2g_{\mu\nu}$ read

$$K^2 - K^{ij}K_{ij} = R(g_{ij}) + 2,$$
$$\nabla^i(K_{ij} - Kg_{ij}) = 0,$$
$$2\rho\,\partial_\rho(K_{ij} - g_{ij}K) + 2K_{ik}K_j^k - 3KK_{ij} + \frac{1}{2}g_{ij}\big[K^{mn}K_{mn} + K^2\big] - g_{ij} = 0, \tag{3.29}$$

where $\nabla_i$ is the covariant derivative with respect to $g_{ij}$. We consider a surface at $\rho = \rho_c$, and relate the bulk and deformed CFT metrics as

$$g_{ij}(\rho_c, x) = \frac{1}{\rho_c}\gamma_{ij}(x). \tag{3.30}$$

For AdS$_3$ the boundary stress tensor defined according to (3.6) (and with the standard choice of $S_{\text{ct}}$ written explicitly below) works out to be

$$T_{ij} = \frac{1}{4G}(K_{ij} - Kg_{ij} + g_{ij}). \tag{3.31}$$

If we use (3.31) to trade $K_{ij}$ for $T_{ij}$ and use $\gamma^{ij}$ to raise indices, it is simple to check that the top line of (3.29) becomes

$$T_i^i = 4G\rho_c \det T_j^i - \frac{1}{8G}R(\gamma), \tag{3.32}$$

which agrees with (3.28) (for a flat boundary metric) under the identification

$$\lambda = \frac{4G\rho_c}{\pi}. \tag{3.33}$$

The boundary stress tensor on the cutoff surface therefore obeys the defining property of a $T\overline{T}$ deformed CFT. For example, this property is enough to fix stress tensor correlation functions at the classical level in the bulk, and so these will agree with the corresponding correlators in the CFT at large $c$, as has been verified explicitly in a few cases [49, 50, 52]. This discussion also makes it clear that the simple relation between a Dirichlet cutoff and the $T\overline{T}$ deformation is lost if bulk matter is included, since the matter stress tensor will show up in the Einstein equation (3.29) leading to modifications of (3.32). This implies that imposing Dirichlet boundary conditions in the presence of matter fields is dual to a more complicated deformation on the CFT side involving nonlocal multitrace operators [49]. One question of interest to us here is whether the simple connection involving pure gravity extends to the quantum level in the bulk. In the latter part of this paper we initiate the quantization procedure and give evidence that the connection holds in perturbation theory in $G \sim 1/c$.

It is also illuminating to understand the bulk description of $T\overline{T}$ by applying the standard AdS/CFT dictionary in the presence of double trace interactions [43]. Here one starts from standard AdS/CFT setup and adds a double trace interaction: $S_{CFT} \to S_{CFT} + \frac{\lambda}{4} \int d^2x \det T^i_j$. Stationarity of the action implies mixed boundary conditions which can be described as fixing a new deformed metric at the conformal boundary of AdS, built out of the original metric and stress tensor. The specific double trace interaction is chosen so that the stress tensor that is conjugate to the deformed metric obeys the trace relation (3.28). It turns out that on-shell the deformed metric is nothing but the induced metric on the $\rho = \rho_c$ slice in the bulk, leading to a derivation of the Dirichlet formulation in pure gravity. In this way of thinking, it seems that the full bulk is really "there" — the surface $\rho = \rho_c$ does not appear as a cutoff but just as a way of thinking about the modified boundary condition. However, for pure gravity the absence of local degrees of freedom implies that there is no sharp distinction between these pictures. Indeed there is a simple way of understanding the relation between them [53]. We start from the bulk action[18]

$$ S = -\frac{1}{16\pi G} \int_M d^3x \sqrt{g}(R+2) - \frac{1}{8\pi G} \int_{\partial M} d^2x \sqrt{g^{(2)}}(K-1) + S_{\text{anom}}. \qquad (3.34) $$

We then consider "integrating out" the region between $\rho = \rho_c$ and the AdS boundary at $\rho = 0$, which is to say that we compute the on-shell action for the enclosed annular spacetime regime. In a theory with local degrees of freedom this would yield a complicated nonlocal expression in terms of data on the two boundary surfaces, but for pure gravity the result is very simple. The answer is simply $S_{\text{ann}} = -\frac{\lambda}{4} \int d^2x \det T^i_j$, where we have written the result in terms of the undeformed metric and stress tensor at the AdS boundary. Coming back to the double trace formulation, we see that the effect of including the double trace interaction is to subtract the action of the annular region, leaving just the action for the region interior to the cutoff surface at $\rho = \rho_c$. It then follows immediately that the double trace and cutoff prescriptions agree, at least at the level of pure gravity in the classical limit.

Let us make a comment on the universality of our starting point, which is the standard two-derivative Einstein-Hilbert action (3.34). A more general action would include higher derivative terms, including those arising from integrating out massive matter fields. However, in three-dimensions, we can perform a field redefinition to put the action back in the form of (3.34).[19] In particular, since the Riemann tensor in 3D may be expressed in terms of the Ricci tensor, a general higher derivative term is a function of the Ricci tensor and its derivatives. Using the field equation of the two-derivative theory, $R_{\mu\nu} + 2g_{\mu\nu} = 0$, the field redefinition $g_{\mu\nu} \to g_{\mu\nu} + \delta g_{\mu\nu}$ changes the two-derivative action as $\delta S \sim \int d^3x \sqrt{g}(R^{\mu\nu} + 2g^{\mu\nu})\delta g_{\mu\nu}$, where we have used $\delta$ to indicate a variation rather than a phase space differential, and so by choosing $\delta g_{\mu\nu}$ appropriately we can generate all the higher derivative terms. This universality of the action (3.34) is the bulk explanation for why correlators of the boundary stress tensor in asymptotically AdS$_3$ space are completely fixed by Virasoro symmetry and the value of the central charge, with no other dependence on the form of the bulk action. This is also the reason why pure gravity in 3D is renormalizable [75]. We expect this universality to hold in our case as well.

---

[18]$S_{\text{anom}}$ is needed to cancel log divergences associated with the Weyl anomaly [74]; see [53] for its explicit form.

[19]This assumes parity invariance, otherwise a gravitational Chern-Simons term should also be included. We also assume that the higher derivative terms can be treated as a perturbation around the Einstein-Hilbert action.

# 4 Warmup: Asymptotic AdS$_3$

This section provides a warmup for the main case of interest. We first recall how to extract a pair of Virasoro algebras from the asymptotic symmetry group of AdS$_3$. This is standard material. We then discuss how to derive the geometric action and symplectic form first discussed by Alekseev and Shatashvili [68]. This was obtained from the Chern-Simons formulation in [26]. Our approach is based on the metric formulation and is easily generalized to the case with a radial cutoff.

## 4.1 Asymptotic symmetry algebra of AdS$_3$

We work with the Bañados metric [59]

$$ds^2 = \frac{d\rho^2}{4\rho^2} + \frac{1}{\rho}(dw + \rho\overline{\mathcal{L}}(\overline{w})d\overline{w})(d\overline{w} + \rho\mathcal{L}(w)dw), \tag{4.1}$$

where $w = \phi + it$ is a coordinate on the cylinder with $\phi \cong \phi + 2\pi$. Global AdS is obtained by taking $\mathcal{L} = \overline{\mathcal{L}} = -\frac{1}{4}$. The boundary stress tensor computed at the $\rho = 0$ boundary is

$$T_{ww} = -\frac{1}{4G}\mathcal{L}, \quad T_{\overline{w}\overline{w}} = -\frac{1}{4G}\overline{\mathcal{L}}, \quad T_{w\overline{w}} = 0. \tag{4.2}$$

The asymptotic Killing vector $\xi$ with components

$$\xi^w = \epsilon(w) - \frac{1}{2}\partial_{\overline{w}}^2\overline{\epsilon}(\overline{w})\rho + \mathcal{O}(\rho^2),$$

$$\xi^{\overline{w}} = \overline{\epsilon}(\overline{w}) - \frac{1}{2}\partial_w^2\epsilon(w)\rho + \mathcal{O}(\rho^2),$$

$$\xi^\rho = \left(\partial_w\epsilon(w) + \partial_{\overline{w}}\overline{\epsilon}(\overline{w})\right)\rho + \mathcal{O}(\rho^2), \tag{4.3}$$

preserves the asymptotic form of the metric in the sense that the variation $\delta_\xi g_{\mu\nu} = \nabla_\mu\xi_\nu + \nabla_\nu\xi_\mu$ does not change the terms in the metric of order $1/\rho$. However the stress tensor components change as

$$\delta_\xi T_{ww} = 2T_{ww}\partial_w\epsilon + \partial_w T_{ww}\epsilon + \frac{1}{8G}\partial_w^3\epsilon,$$

$$\delta_\xi T_{\overline{w}\overline{w}} = 2T_{\overline{w}\overline{w}}\partial_{\overline{w}}\overline{\epsilon} + \partial_{\overline{w}} T_{\overline{w}\overline{w}}\overline{\epsilon} + \frac{1}{8G}\partial_{\overline{w}}^3\overline{\epsilon}. \tag{4.4}$$

The charges (3.26) are

$$Q[\xi] = \frac{i}{2\pi}\int_0^{2\pi} T_{ti}\xi^i d\phi. \tag{4.5}$$

We separate out the parts proportional to $\epsilon$ and $\overline{\epsilon}$,

$$Q[\epsilon] = -\frac{1}{2\pi}\int_0^{2\pi} T_{ww}\epsilon d\phi, \quad \overline{Q}[\overline{\epsilon}] = \frac{1}{2\pi}\int_0^{2\pi} T_{\overline{w}\overline{w}}\overline{\epsilon}d\phi. \tag{4.6}$$

Since we established the relation (3.25) we can extract the Poisson brackets of the charges using (2.11), which here reads

$$\{Q[\epsilon_1], Q[\epsilon_2]\} = -\delta_{\xi_1}Q[\epsilon_2], \quad \{\overline{Q}[\overline{\epsilon}_1], \overline{Q}[\overline{\epsilon}_2]\} = -\delta_{\xi_1}\overline{Q}[\overline{\epsilon}_2], \tag{4.7}$$

with the mixed bracket vanishing. This gives

$$\{Q[\epsilon_1], Q[\epsilon_2]\} = -\frac{1}{2\pi}\int_0^{2\pi}\left[(\partial_w\epsilon_1\epsilon_2 - \epsilon_1\partial_w\epsilon_2)T_{ww} + \frac{c}{24}(\partial_w^3\epsilon_1\epsilon_2 - \epsilon_1\partial_w^3\epsilon_2)\right]d\phi \qquad (4.8)$$

and similarly for $\{\overline{Q}[\overline{\epsilon}_1], \overline{Q}[\overline{\epsilon}_2]\}$, and where we used the Brown-Henneaux formula $c = \frac{3}{2G}$.

We now write the Fourier expansion[20]

$$T_{ww}(w) = -\sum_m Q_m e^{imw},$$
$$T_{\overline{w}\overline{w}}(\overline{w}) = -\sum_m \overline{Q}_m e^{-im\overline{w}}, \qquad (4.9)$$

i.e.

$$Q_m = Q[e^{-imw}], \quad \overline{Q}_m = \overline{Q}[-e^{im\overline{w}}]. \qquad (4.10)$$

(4.8) then gives the Virasoro algebra[21]

$$i\{Q_m, Q_n\} = (m-n)Q_{m+n} + \frac{c}{12}m^3\delta_{m,-n}, \qquad (4.11)$$

along with the same formula with $Q$'s replaced by $\overline{Q}$'s.

## 4.2 The Alekseev-Shatashvili symplectic form

We now wish to use (3.25) to extract a useful expression for $\Omega$. The phase space of interest is not the full space of gravity solutions with specified boundary conditions, but rather a single coadjoint orbit, which here refers to the space of all stress tensors that can be obtained by some diffeomorphism transformation starting from global AdS. On general grounds, this space is a symplectic manifold and the symplectic form may be obtained as a particular case of a more general construction due to Kirillov and Kostant [77]. Rather than going through the details of this construction, we can obtain the result of interest in a way that follows easily from what we have so far. Focusing on the holomorphic stress tensor, we recall that global AdS corresponds to $T_{ww} = \frac{c}{24}$. We then perform a finite diffeomorphism, which on the boundary at fixed time acts as $\phi \to f(\phi)$. This can be carried out explicitly in the bulk, but all we need is the familiar statement that the stress tensor transforms as a tensor plus a Schwarzian term, and so we obtain

$$T_{ww} = \frac{c}{12}\left(\frac{1}{2}f'^2 + \{f(\phi), \phi\}\right), \qquad (4.12)$$

with

$$\{f(\phi), \phi\} = \frac{f'''}{f'} - \frac{3}{2}\frac{f''^2}{f'^2}. \qquad (4.13)$$

There is a gauge redundancy is passing from the space of stress tensors to the space of functions $f(\phi)$ since the starting point $T_{ww} = \frac{c}{24}$ is invariant under an PSL$(2,R)$ group of reparameterizations that maps

$$\tan\left(\frac{f}{2}\right) \to \frac{a\tan\left(\frac{f}{2}\right) + b}{c\tan\left(\frac{f}{2}\right) + d}. \qquad (4.14)$$

---

[20]The minus signs are included for agreement with standard CFT conventions, where they originate from the conformal transformation from the plane to the cylinder. See for example [76].

[21]The shifted modes $L_m = Q_m + \frac{c}{24}\delta_{m,0}$ obey the more familiar $i\{L_m, L_n\} = (m-n)L_{m+n} + \frac{c}{12}m(m^2-1)\delta_{m,-n}$.

Indeed, (4.12) is just the same as $c/12\{\tan(f/2),\phi\}$, so this result follows from the invariance of the Schwarzian derivative under $PSL(2,R)$. Therefore, the phase space is really the coset $\text{diff}(S^1)/PSL(2,R)$ [69]. We want to compute the symplectic form on this space. To clarify, we could in principle write the symplectic form in terms of the stress tensor but the result would be nonlocal (one essentially needs to invert (4.12)), while the result in terms of $f$ is local and relatively simple.

Expressed in terms of $f$ the charges are

$$Q[\epsilon] = -\frac{c}{24\pi}\int_0^{2\pi} d\phi \left(\frac{1}{2}f'^2 + \frac{f'''}{f'} - \frac{3}{2}\frac{f''^2}{f'^2}\right)\epsilon. \tag{4.15}$$

To extract $\Omega$ it will be convenient to contract with a vector field and write (3.25) as

$$i_{V_{\epsilon_1}} i_{V_{\epsilon_2}} \Omega = -i_{V_{\epsilon_1}} \delta Q[\epsilon_2]. \tag{4.16}$$

By considering the composition of the reparameterization $\phi \to f(\phi)$ with an infinitesimal reparameterization $\phi \to \phi + \epsilon(\phi)$ we deduce

$$\delta_\epsilon f \equiv i_{V_\epsilon} \delta f = f'\epsilon. \tag{4.17}$$

We now compute the right hand side of (4.16) and integrate by parts to put the result in a form that is manifestly antisymmetric under $\epsilon_1 \leftrightarrow \epsilon_2$. We find

$$i_{V_{\epsilon_1}} \delta Q[\epsilon_2] = -\frac{c}{24\pi}\int_0^{2\pi} d\phi \left[\left(\frac{1}{2}f'^2 + \{f(\phi),\phi\}\right)(\epsilon_1'\epsilon_2 - \epsilon_1\epsilon_2') + \frac{1}{2}(\epsilon_1'''\epsilon_2 - \epsilon_1\epsilon_2''')\right]. \tag{4.18}$$

Using (4.17) it is now simple to solve for $\Omega$ as

$$\Omega = -\frac{c}{24\pi}\int_0^{2\pi} d\phi \left[\left(\frac{1}{2}f'^2 + \{f(\phi),\phi\}\right)\left(\frac{\delta f}{f'}\right)' \wedge \frac{\delta f}{f'} + \frac{1}{2}\left(\frac{\delta f}{f'}\right)''' \wedge \frac{\delta f}{f'}\right]. \tag{4.19}$$

After integrating by parts this can be simplified to

$$\Omega = -\frac{c}{48\pi}\int_0^{2\pi} d\phi \left(\frac{\delta f' \wedge \delta f''}{f'^2} - \delta f \wedge \delta f'\right). \tag{4.20}$$

We further note

$$\Omega = \delta\Upsilon, \tag{4.21}$$

with

$$\Upsilon = \frac{c}{48\pi}\int_0^{2\pi} d\phi \left(\frac{\delta f''}{f'} + f\delta f'\right). \tag{4.22}$$

We also have the obvious analogous expressions on the anti-holomorphic side.

These formulas can be further simplified by defining (e.g. [26])

$$F = if + \ln f' \tag{4.23}$$

in terms of which

$$T_{ww} = -\frac{c}{12}\left(\frac{1}{2}F'^2 - F''\right),$$

$$\Omega = -\frac{c}{48\pi}\int_0^{2\pi} d\phi\, \delta F \wedge \delta F',$$

$$\Upsilon = -\frac{c}{48\pi}\int_0^{2\pi} d\phi\, F \delta F'. \tag{4.24}$$

These expressions are simply the stress tensor and symplectic form for a free boson whose stress tensor includes a linear dilaton (or background charge) contribution. One should however recall the PSL(2,R) gauge symmetry. In this form it is clear that quantization yields a Hilbert space that corresponds to the Virasoro vacuum module.

In the above we started from the vacuum stress tensor value $T_{ww} = \frac{c}{24}$, but it is simple to generalize to other values. Starting from $T_{ww} = \frac{c}{24\pi}\kappa$ the only change is that we multiply the second term on the right hand of (4.20) by $\kappa$. For generic values of $\kappa$ the SL(2,R) gauge symmetry is now $U(1)$. The enhancement to SL(2,R) occurs for $\kappa = \frac{1}{n^2}$. The $n > 1$ cases correspond to conical defect solutions in the bulk [26].

Given an expression for the symplectic potential $\Upsilon$ and the Hamiltonian $H$, we can immediately write down a phase space action $S = \int(\Upsilon - H dt)$. By design, the corresponding phase space path integral computes Virasoro characters [68, 78].

### 4.3 Perturbative expansion

We now write out the above formulas in a form that facilitates comparison to the results we will derive at finite cutoff $\rho_c$. We write the coordinate transformation in the form

$$w \to f(w) = w + A(\phi, t) + iB(\phi, t), \quad \overline{w} \to \overline{f}(\overline{w}) = \overline{w} + A(\phi, t) - iB(\phi, t) \tag{4.25}$$

corresponding to

$$\phi \to \phi + A(\phi, t), \quad t \to t + B(\phi, t). \tag{4.26}$$

We note that $A+iB$ ($A-iB$) is necessarily (anti-)holomorphic by definition in (4.25). Expanding in powers of $A$ and $B$ we have to cubic order,

$$
\begin{aligned}
T_{tt} &= -(T_{ww} + T_{\overline{w}\overline{w}}) \\
&= \frac{c}{12}\Big(-1 - 2A' - 2A''' + 3A''^2 - 3B''^2 + 2A'A''' - 2B'B''' - A'^2 + B'^2 \\
&\quad - 6A'A''^2 + 12A''B'B'' + 6A'B''^2 - 2A'^2A''' + 2A'''B'^2 + 4A'B'B'''\Big) + \dots, \\
T_{\phi t} &= i(T_{ww} - T_{\overline{w}\overline{w}}) \\
&= \frac{c}{6}\Big(-B' - B''' + 3A''B'' + A'''B' + A'B''' - A'B' \\
&\quad - 3A''^2B' - 6A'A''B'' + 3B'B''^2 - 2A'A'''B' - A'^2B''' + B'^2B'''\Big) + \dots, \tag{4.27}
\end{aligned}
$$

where now $' = \frac{\partial}{\partial \phi}$. Tracelessness implies $T_{\phi\phi} = -T_{tt}$. The symplectic form can be read off from

$$
\begin{aligned}
\Upsilon = -\frac{ic}{24\pi}\int d\phi\, \Big(&\big[B' + (B' - 2A'B' + 3A'^2B' - B'^3)''\big]\delta A \\
&+ \big[1 + A' + (A' + B'^2 - A'^2 - 3A'B'^2 + A'^3)''\big]\delta B + \dots\Big). \tag{4.28}
\end{aligned}
$$

# 5 Cutoff geometries and their charge algebra

## 5.1 Metrics obeying Dirichlet boundary condition

We now discuss the general three-dimensional gravity solutions with negative cosmological constant whose boundary is a finite cylinder. Specifically, we impose $ds^2|_{\rho=\rho_c} = \frac{1}{\rho_c} dw d\overline{w}$ where $w = \phi + it$ and $\phi \cong \phi + 2\pi$. Such solutions can be obtained starting from (4.1), which we now write in primed coordinates,

$$ds^2 = \frac{d\rho'^2}{4\rho'^2} + \frac{1}{\rho'}(dw + \rho'\overline{\mathcal{L}}(\overline{w}')d\overline{w}')(d\overline{w}' + \rho'\mathcal{L}(w')dw'). \tag{5.1}$$

At this stage, we temporarily relax any periodicity requirements on the functions $\mathcal{L}(w)$ and $\overline{\mathcal{L}}(\overline{w})$), and instead allow them to be arbitrary. We then define new coordinates $(w,\overline{w})$ via[22]

$$dw = dw' + \rho_c\overline{\mathcal{L}}(\overline{w}')d\overline{w}',$$
$$d\overline{w} = d\overline{w}' + \rho_c\mathcal{L}(w')dw', \tag{5.2}$$

along with $\rho = \rho'$. With this we arrive at a metric satisfying the desired boundary conditions

$$ds^2 = \frac{d\rho^2}{4\rho^2} + \frac{1}{\rho}\frac{\left[(1-\rho\rho_c\mathcal{L}\overline{\mathcal{L}})dw + (\rho-\rho_c)\overline{\mathcal{L}}d\overline{w}\right]\left[(1-\rho\rho_c\mathcal{L}\overline{\mathcal{L}})d\overline{w} + (\rho-\rho_c)\mathcal{L}dw\right]}{(1-\rho_c^2\mathcal{L}\overline{\mathcal{L}})^2}. \tag{5.3}$$

This is a solution provided the functions $\mathcal{L} = \mathcal{L}(w,\overline{w})$ and $\overline{\mathcal{L}}(w,\overline{w})$ obey

$$\partial_{\overline{w}}\mathcal{L} = -\rho_c\overline{\mathcal{L}}\partial_w\mathcal{L}$$
$$\partial_w\overline{\mathcal{L}} = -\rho_c\mathcal{L}\partial_{\overline{w}}\overline{\mathcal{L}}, \tag{5.4}$$

which are of course equivalent to the statements that these functions are (anti)holomorphic with respect to the primed coordinates. Recalling that $w = \phi + it$ with $\phi \cong \phi + 2\pi$, we now also write $\mathcal{L} = \mathcal{L}(\phi,t)$ and $\overline{\mathcal{L}} = \overline{\mathcal{L}}(\phi,t)$, and demand that these functions respect the $\phi$ periodicity. If we rewrite (5.4) as

$$-i\partial_t\mathcal{L} = \frac{1+\rho_c\overline{\mathcal{L}}}{1-\rho_c\overline{\mathcal{L}}}\partial_\phi\mathcal{L},$$
$$i\partial_t\overline{\mathcal{L}} = \frac{1+\rho_c\mathcal{L}}{1-\rho_c\mathcal{L}}\partial_\phi\overline{\mathcal{L}}, \tag{5.5}$$

we see that we can start with some $t = t_0$ initial data $(\mathcal{L}(\phi,t_0),\overline{\mathcal{L}}(\phi,t_0))$ that respects the $\phi$ periodicity, and that solving (5.5) will preserve the periodicity. Hence the space of solutions is labelled by two arbitrary periodic functions of $\phi$ giving the initial conditions.

The stress tensor evaluated at the cutoff surface using (3.31) is

$$T_{ww} = -\frac{1}{4G}\frac{\mathcal{L}}{1-\rho_c^2\mathcal{L}\overline{\mathcal{L}}}, \quad T_{\overline{w}\overline{w}} = -\frac{1}{4G}\frac{\overline{\mathcal{L}}}{1-\rho_c^2\mathcal{L}\overline{\mathcal{L}}}, \quad T_{w\overline{w}} = -\frac{\rho_c}{4G}\frac{\mathcal{L}\overline{\mathcal{L}}}{1-\rho_c^2\mathcal{L}\overline{\mathcal{L}}}. \tag{5.6}$$

We verify that the cutoff stress tensor obeys the $T\overline{T}$ trace relation

$$T_{w\overline{w}} = -4G\rho_c\Big(T_{ww}T_{\overline{w}\overline{w}} - (T_{w\overline{w}})^2\Big). \tag{5.7}$$

---

[22]Since at the moment we are not imposing any periodicity requirement there is no obstacle to integrating (5.2) to obtain $(w,\overline{w})$.

## 5.2 Boundary preserving vector fields

Given a solution of the form (5.3) we define a "boundary preserving vector" $\xi$ to be a smooth vector field such that $\delta_\xi g_{\mu\nu} = \nabla_\mu \xi_\nu + \nabla_\nu \xi_\mu$ obeys $\delta_\xi g_{\mu\nu}\big|_{\rho=\rho_c} = 0$. That is, it leaves all components of the metric at the boundary invariant. We showed in Section 3.2 that such vector fields will lead to charges (3.26).

Due to the nature of our problem we can perform a local analysis near the boundary. We start with the ansatz

$$
\begin{aligned}
\xi^w(\rho, w, \overline{w}) &= f^w(w, \overline{w}) + \mathcal{O}(\rho - \rho_c), \\
\xi^{\overline{w}}(\rho, w, \overline{w}) &= f^{\overline{w}}(w, \overline{w}) + \mathcal{O}(\rho - \rho_c), \\
\xi^\rho(\rho, w, \overline{w}) &= \rho f^\rho(w, \overline{w}).
\end{aligned}
\tag{5.8}
$$

Respecting the boundary conditions requires

$$
\begin{aligned}
\partial_{\overline{w}} f^w &= -\frac{\rho_c \overline{\mathcal{L}}}{1 + \rho_c^2 \mathcal{L}\overline{\mathcal{L}}}(\partial_w f^w + \partial_{\overline{w}} f^{\overline{w}}), \\
\partial_w f^{\overline{w}} &= -\frac{\rho_c \mathcal{L}}{1 + \rho_c^2 \mathcal{L}\overline{\mathcal{L}}}(\partial_w f^w + \partial_{\overline{w}} f^{\overline{w}}),
\end{aligned}
\tag{5.9}
$$

and we also find that $f^\rho$ is determined as

$$
f^\rho = \left(\frac{1 - \rho_c^2 \mathcal{L}\overline{\mathcal{L}}}{1 + \rho_c^2 \mathcal{L}\overline{\mathcal{L}}}\right)(\partial_w f^w + \partial_{\overline{w}} f^{\overline{w}}).
\tag{5.10}
$$

This is not yet sufficient, and we also need to add $\mathcal{O}(\rho - \rho_c)$ terms to $\xi^{w,\overline{w}}$. The final result is then

$$
\begin{aligned}
\xi^w(\rho, w, \overline{w}) &= f^w(w, \overline{w}) - \frac{1}{2}\partial_{\overline{w}} f^\rho(w, \overline{w})(\rho - \rho_c) + O\big((\rho - \rho_c)^2\big), \\
\xi^{\overline{w}}(\rho, w, \overline{w}) &= f^{\overline{w}}(w, \overline{w}) - \frac{1}{2}\partial_w f^\rho(w, \overline{w})(\rho - \rho_c) + O\big((\rho - \rho_c)^2\big), \\
\xi^\rho(\rho, w, \overline{w}) &= \rho f^\rho(w, \overline{w}).
\end{aligned}
\tag{5.11}
$$

If we wished to fix radial gauge, $g_{\rho i} = 0$ away from the boundary we would need to determine the full extension of $\xi$ into the bulk. However, this is not necessary since the details of this extension are "pure gauge" and do not affect the boundary charges or their transformations. Related to this, working at the above order is sufficient to allow us to use (3.31) with $K_{ij} = -\rho \partial_\rho g_{ij}$ to compute the boundary stress tensor, since the violation of radial gauge away from the boundary has vanishing effect on $T_{ij}$ once we set $\rho = \rho_c$.

In Section 5.1 we emphasized that bulk solutions could be labelled by a pair of free periodic functions on the boundary circle at some initial time $t_0$. We can parameterize the boundary preserving vector field in a similar fashion. This is important, because this construction will render its components tangent to the boundary field independent on the boundary circle at $t = t_0$ even though the full vector field is field dependent. This property will be crucial when computing the charge algebra. To this end, we rewrite (5.9) as

$$
\begin{aligned}
-i\partial_t f^w &= \partial_\phi f^w + \frac{2\rho_c \overline{\mathcal{L}}}{(1 - \rho_c \mathcal{L})(1 - \rho_c \overline{\mathcal{L}})}(\partial_\phi f^w + \partial_\phi f^{\overline{w}}), \\
i\partial_t f^{\overline{w}} &= \partial_\phi f^{\overline{w}} + \frac{2\rho_c \mathcal{L}}{(1 - \rho_c \mathcal{L})(1 - \rho_c \overline{\mathcal{L}})}(\partial_\phi f^w + \partial_\phi f^{\overline{w}}).
\end{aligned}
\tag{5.12}
$$

We then write down fixed initial data for $(f^w, f^{\overline{w}})$ on the same $t = t_0$ surface which we use to define the charges. This ensures that when we compute $\delta Q[\xi]$ we can take $\delta \xi^i = 0$.

It is worth comparing this result with the analysis in [43] and [44]. The relation with the former follows most straightforwardly by matching (5.11) with equations (3.34) in [43] at $\rho = \rho_c$, giving $f^w(w, \overline{w}) = f(w') - \rho_c \bar{\mathcal{L}}_{\bar{f}}(\overline{w}')$ and its right-moving conjugate. It was noted that the functions $\mathcal{L}_f$ have nonzero winding around the spatial circle even when $f$ doesn't, an issue which was studied in detail in [44]. Here, we find a different way around the issue, by using the functions $(f^w, f^{\overline{w}})$ as a basis to label the diffeomorphisms. These functions do not have nonzero winding, as discussed in the previous paragraph, and we will find a different Poisson bracket algebra for the charges in this basis.

The relation with [44] can also be clarified at this point. As explained in this section, we will label our changes by specifying the functions $f^w(\phi)$ and $f^{\overline{w}}(\phi)$ on the surface $t = t_0$. These are fixed functions of a state-independent coordinate. In [44], however, the charges were labeled by fixed functions $f$ and $\bar{f}$ of state-dependent coordinates (coinciding with $(w', \overline{w}')$ in (5.1)). These necessarily have nonzero variations, for example $\{\partial_\phi f, \cdot\} = f'\{\frac{\mathcal{H} - \mathcal{P}}{R_u}, \cdot\}$ in the notation of that paper. If we were to redo the calculation [44] without the contributions from these Poisson brackets, we would find the classical part[23] of the result we will obtain in (5.29).

## 5.3 Boundary charges

As in (4.5) the boundary charge (3.26) associated to a boundary preserving vector field $\xi$ is defined to be

$$Q[\xi] = \frac{i}{2\pi} \int_0^{2\pi} T_{ti} \xi^i d\phi \,. \tag{5.13}$$

As noted in the previous section, we are free to specify vector fields

$$\xi_n|_{t=t_0} = e^{-in\phi} \partial_w, \quad \overline{\xi}_n|_{t=t_0} = -e^{in\phi} \partial_{\overline{w}} \tag{5.14}$$

on the $t = t_0$ surface. This suggests generalizing the Virasoro charges (4.10) according to (5.13) and (5.6) by

$$Q_n = Q[e^{-in\phi} \partial_w] = \frac{1}{8\pi G} \int d\phi \frac{(1 - \rho_c \overline{\mathcal{L}})\mathcal{L}}{1 - \rho_c^2 \mathcal{L}\overline{\mathcal{L}}} e^{-in\phi} \,,$$

$$\overline{Q}_n = Q[-e^{in\phi} \partial_{\overline{w}}] = \frac{1}{8\pi G} \int d\phi \frac{(1 - \rho_c \mathcal{L})\overline{\mathcal{L}}}{1 - \rho_c^2 \mathcal{L}\overline{\mathcal{L}}} e^{in\phi} \,, \tag{5.15}$$

which we have evaluated on the $t = t_0$ surface. The energy and momentum charges are now

$$H = Q[-i\partial_t] = Q_0 + \overline{Q}_0 \,,$$

$$P = Q[\partial_\phi] = Q_0 - \overline{Q}_0 \,. \tag{5.16}$$

It will be useful to compute the bracket of these charges with the Hamiltonian indirectly here so we can later check that it is reproduced by our direct calculation of the full Poisson algebra. Recalling (2.9), we see that the charges can have both an explicit time dependence in their definition and a time dependence due to the Hamiltonian's flow on the phase space. The only possible source for an explicit time dependence in the charges $Q[\xi_n]$ and $\overline{Q}[\overline{\xi}_n]$ would be the from the components of $\xi_n$ and $\overline{\xi}_n$. Indeed, these will have non-trivial time dependence

---

[23]With "classical part" we mean the part where $\ell$ is reintroduced in (5.29) and then set to 0. For the undeformed CFT, this removes the central charge of the algebra, which is classically not accessible.

implied by the evolution equations (5.12). However, we see in (2.9) that if we calculate the time derivative ignoring any explicit time dependence, the result must be the Poisson bracket with the Hamiltonian. Using (5.5), we take the time derivative of the expression in (5.15) and integrate by parts to get

$$\partial_t Q_n = -n Q_n - \frac{n}{4\pi G}\int d\phi\, \frac{\rho_c \mathcal{L}\overline{\mathcal{L}}}{1-\rho_c^2 \mathcal{L}\overline{\mathcal{L}}}e^{-in\phi}$$

$$\partial_t \overline{Q}_n = -n \overline{Q}_n - \frac{n}{4\pi G}\int d\phi\, \frac{\rho_c \mathcal{L}\overline{\mathcal{L}}}{1-\rho_c^2 \mathcal{L}\overline{\mathcal{L}}}e^{in\phi}\,. \tag{5.17}$$

We now make some remarks regarding the (non)conservation of these charges. We first of all note that energy and momentum are conserved, $\partial_t H = \partial_t P = 0$, following from the fact that the boundary metric is invariant under translations in $t$ and $\phi$. Furthermore, we see from (5.12) that $\xi_0$ and $\overline{\xi}_0$ are constants in time, so the total time derivatives of $H$ and $P$ vanish in addition to their bracket with the Hamiltonian. The $n \neq 0$ charges are not conserved. In the asymptotically AdS case with $\rho_c = 0$, we can of course define conserved charges by introducing explicit time dependence, $Q_n e^{nt}$ and $\overline{Q}_n e^{nt}$. At nonzero $\rho_c$ no such simple construction is available.

## 5.4 Variation of the stress tensor

We now wish to compute the variation of the stress tensor under our boundary preserving vectors. We proceed by computing $(\delta_\xi \mathcal{L}, \delta_\xi \overline{\mathcal{L}})$ and then by using (5.6). To extract these variations we first evaluate $\delta_\xi \partial_\rho g_{ij}\big|_{\rho_c}$ and then ask what $(\delta_\xi \mathcal{L}, \delta_\xi \overline{\mathcal{L}})$ reproduces this. Note that the problem is over constrained since we have two free parameters but three equations. The existence of a solution requires that the following consistency condition holds

$$(1+\rho_c^2 \mathcal{L}\overline{\mathcal{L}})\partial_w \partial_{\overline{w}} f^\rho + \rho_c(\overline{\mathcal{L}}\partial_w^2 f^\rho + \mathcal{L}\partial_{\overline{w}}^2 f^\rho) = 0\,. \tag{5.18}$$

With some work, by using (5.4) and (5.9) this can indeed be shown to hold. We then find

$$\delta_\xi \mathcal{L} = (f^w - \rho_c \overline{\mathcal{L}}f^{\overline{w}})\partial_w \mathcal{L} + \frac{2}{1+\rho_c^2 \mathcal{L}\overline{\mathcal{L}}}(\partial_w f^w - \rho_c^2 \mathcal{L}\overline{\mathcal{L}}\partial_{\overline{w}}f^{\overline{w}})\mathcal{L}$$

$$+ \frac{1-\rho_c^2 \mathcal{L}\overline{\mathcal{L}}}{2(1+\rho_c^2 \mathcal{L}\overline{\mathcal{L}})}(\rho_c^2 \mathcal{L}^2 \partial_{\overline{w}}^2 f^\rho - \partial_w^2 f^\rho)\,, \tag{5.19}$$

as well as the same formula with barred and unbarred quantities exchanged. Plugging into (5.6) gives the transformation of the stress tensor, but we refrain from writing this out.

## 5.5 Charge algebra

The Poisson bracket algebra among the charge $Q[\xi]$ may be extracted from the transformation of the charges under an asymptotic symmetry, the master formula being $\{Q[\xi_1], Q[\xi_2]\} = -\delta_{V_{\xi_1}}Q[\xi_2]$, as in (2.11). We use the basis of charges $(Q_n, \overline{Q}_n)$ defined in (5.15), and we should use the same basis for our boundary preserving vectors. We use $(\delta_m, \overline{\delta}_m)$ to denote variations with respect to boundary preserving vectors $\xi_n$ and $\overline{\xi}_n$ introduced in (5.14). That is, we have[24]

$$\delta_m: \quad f^w = e^{-im\phi}\,, \quad f^{\overline{w}} = 0\,,$$

$$\overline{\delta}_m: \quad f^w = 0\,, \qquad f^{\overline{w}} = -e^{im\phi}\,. \tag{5.20}$$

---

[24]We are now working on the $t = t_0$ surface.

In this notation we have

$$\{Q_m, Q_n\} = -\overline{\delta}_m Q_n, \quad \{\overline{Q}_m, \overline{Q}_n\} = -\overline{\delta}_m \overline{Q}_n, \quad \{Q_m, \overline{Q}_n\} = -\delta_m \overline{Q}_n. \tag{5.21}$$

To arrive at the charge algebra we need to compute these variations using (5.19) and then express the result in terms of the charges.

However, we first need to reexpress (5.19) in terms of our initial data on the $t = 0$ surface, which in particular means using (5.5) and (5.12) to trade away all $t$ derivatives in favor of $\phi$ derivatives. This is straightforward but somewhat messy and we do not write out the explicit result for $(\delta_\xi \mathcal{L}, \delta_\xi \overline{\mathcal{L}})$ in this form. Taking these expressions and using them to evaluate the variations in (5.21) we encounter a nice feature: the integrands are exponentials multiplying a function of $(\mathcal{L}, \overline{\mathcal{L}})$ and their $\phi$ derivatives, but it turns out that all $\phi$ derivatives appear as total derivatives. We therefore integrate by parts and take the $\phi$ derivatives to act on the exponentials, where they simply generate polynomials in $m$ and $n$. Again, the algebra is a bit messy, but easy enough to carry out on the computer. We arrive at the following Poisson brackets:

$$i\{Q_m, Q_n\} = \frac{1}{8\pi G} \int_0^{2\pi} d\phi\, e^{-i(m+n)\phi} \left\{ \frac{1}{2} m^3 + (m-n) \frac{\mathcal{L}}{1 - \rho_c^2 \mathcal{L}\overline{\mathcal{L}}} \right.$$
$$\left. - \frac{1}{2} mn(m-n)\rho_c \frac{\overline{\mathcal{L}} - \frac{1}{2}\rho_c \mathcal{L}^2 - \frac{1}{2}\rho_c \overline{\mathcal{L}}^2 + \rho_c^2 \mathcal{L}\overline{\mathcal{L}}^2 + 2\rho_c^2 \mathcal{L}^2 \overline{\mathcal{L}} - 2\rho_c \mathcal{L}\overline{\mathcal{L}} - \rho_c^3 \mathcal{L}^2 \overline{\mathcal{L}}^2}{(1 - \rho_c \mathcal{L})^2 (1 - \rho_c \overline{\mathcal{L}})^2} \right\}, \tag{5.22}$$

$$i\{\overline{Q}_m, \overline{Q}_n\} = \frac{1}{8\pi G} \int_0^{2\pi} d\phi\, e^{i(m+n)\phi} \left\{ \frac{1}{2} m^3 + (m-n) \frac{\overline{\mathcal{L}}}{1 - \rho_c^2 \mathcal{L}\overline{\mathcal{L}}} \right.$$
$$\left. - \frac{1}{2} mn(m-n)\rho_c \frac{\mathcal{L} - \frac{1}{2}\rho_c \mathcal{L}^2 - \frac{1}{2}\rho_c \overline{\mathcal{L}}^2 + \rho_c^2 \mathcal{L}^2 \overline{\mathcal{L}} + 2\rho_c^2 \mathcal{L}\overline{\mathcal{L}}^2 - 2\rho_c \mathcal{L}\overline{\mathcal{L}} - \rho_c^3 \mathcal{L}^2 \overline{\mathcal{L}}^2}{(1 - \rho_c \mathcal{L})^2 (1 - \rho_c \overline{\mathcal{L}})^2} \right\}, \tag{5.23}$$

$$i\{Q_m, \overline{Q}_n\} = \frac{1}{8\pi G} \int_0^{2\pi} d\phi\, e^{-i(m-n)\phi} \left\{ -\frac{mn(m-n+2n\rho_c\overline{\mathcal{L}})}{4(1 - \rho_c\overline{\mathcal{L}})^2} - \frac{mn(m-n-2m\rho_c\mathcal{L})}{4(1 - \rho_c\mathcal{L})^2} \right.$$
$$\left. + \frac{(m-n)\rho_c \mathcal{L}\overline{\mathcal{L}}}{1 - \rho_c^2 \mathcal{L}\overline{\mathcal{L}}} \right\}. \tag{5.24}$$

A sensitive check of the intermediate steps is that $\{Q_m, Q_n\}$ and $\{\overline{Q}_m, \overline{Q}_n\}$ are found to be anti-symmetric under $m \leftrightarrow n$.

The Poisson brackets involving the Hamiltonian $H = Q_0 + \overline{Q}_0$ and momentum $P = Q_0 - \overline{Q}_0$ are relatively simple,

$$i\{H, Q_n\} = -nQ_n - \frac{n}{4\pi G} \int_0^{2\pi} d\phi\, \frac{\rho_c \mathcal{L}\overline{\mathcal{L}}}{1 - \rho_c^2 \mathcal{L}\overline{\mathcal{L}}} e^{-in\phi},$$
$$i\{H, \overline{Q}_n\} = -n\overline{Q}_n - \frac{n}{4\pi G} \int_0^{2\pi} d\phi\, \frac{\rho_c \mathcal{L}\overline{\mathcal{L}}}{1 - \rho_c^2 \mathcal{L}\overline{\mathcal{L}}} e^{in\phi}, \tag{5.25}$$

$$i\{P, Q_n\} = -nQ_n,$$
$$i\{P, \overline{Q}_n\} = n\overline{Q}_n. \tag{5.26}$$

The fact that the Poisson brackets involving $P$ have no $\rho_c$ dependence implies, upon replacing Poisson brackets by commutators, momentum quantization. On the other hand, the $\rho_c$ dependence in the Poisson brackets involving $H$ is expected given that the energy spectrum of a $T\overline{T}$ deformed theory is modified. As expected, the brackets with the Hamiltonian match our earlier indirect calculation (5.17).

To complete the story we need to use (5.15) to trade away $(\mathcal{L}, \overline{\mathcal{L}})$ for $(Q_n, \overline{Q}_n)$. It is convenient to define

$$q(\phi) = 4G \sum_n Q_n e^{in\phi}, \quad \overline{q}(\phi) = 4G \sum_n \overline{Q}_n e^{-in\phi}, \tag{5.27}$$

so that

$$\mathcal{L}(\phi) = \frac{q(\phi)}{1 - \rho_c \overline{q}(\phi)}, \quad \overline{\mathcal{L}}(\phi) = \frac{\overline{q}(\phi)}{1 - \rho_c q(\phi)}. \tag{5.28}$$

This gives [25]

$$i\{Q_m, Q_n\} = \frac{1}{8\pi G} \int_0^{2\pi} d\phi\, e^{-i(m+n)\phi} \left\{ \frac{1}{2} m^3 + (m-n) \frac{(1-\rho_c q)q}{1-\rho_c(q+\overline{q})} \right.$$
$$\left. - \frac{1}{2} mn(m-n)\rho_c \frac{\overline{q} - \frac{1}{2}\rho_c(q+\overline{q})^2}{(1-\rho_c(q+\overline{q}))^2} \right\},$$

$$i\{\overline{Q}_m, \overline{Q}_n\} = \frac{1}{8\pi G} \int_0^{2\pi} d\phi\, e^{i(m+n)\phi} \left\{ \frac{1}{2} m^3 + (m-n) \frac{(1-\rho_c \overline{q})\overline{q}}{1-\rho_c(q+\overline{q})} \right.$$
$$\left. - \frac{1}{2} mn(m-n)\rho_c \frac{q - \frac{1}{2}\rho_c(q+\overline{q})^2}{(1-\rho_c(q+\overline{q}))^2} \right\}, \tag{5.29}$$

$$i\{Q_m, \overline{Q}_n\} = \frac{1}{8\pi G} \int_0^{2\pi} d\phi\, e^{i(n-m)\phi} \left\{ \frac{\rho_c(m-n)q\overline{q}}{1-\rho_c(q+\overline{q})} + \frac{1}{2}\rho_c\, mn\, \frac{nq - m\overline{q} + \frac{\rho_c}{2}(m-n)(q+\overline{q})^2}{(1-\rho_c(q+\overline{q}))^2} \right\}.$$

We emphasize that this result has been derived without assuming any details about the background spacetime other than its finite cylinder boundary. In the limit $\rho_c \to 0$ the right-hand side becomes linear in the functions $(q, \overline{q})$ that label the state. It reduces to the asymptotic symmetry algebra (4.11) of asymptotically AdS$_3$ spacetimes. In the next section, we will analyze this result perturbatively around global AdS$_3$.

Let us also comment on the range of validity of this charge algebra. The starting point was the usual Einstein-Hilbert action. However, as discussed in Section 3.3, adding higher derivative terms will not change the story, since in 3D these can be removed by a field redefinition. Furthermore, if other matter fields are present we can imagine integrating them out. As long as they are massive and have sufficiently local interactions such that this procedure results in a series of higher derivative terms for the metric, our reasoning should go through.

It is sometimes useful to rewrite this result in terms of the combinations

$$P_m = Q_m + \overline{Q}_{-m}, \qquad\qquad J_m = Q_m - \overline{Q}_{-m}, \tag{5.30}$$

and similarly $p = q + \overline{q}$, $j = q - \overline{q}$. Indeed, the Poisson bracket for $P_m$ with itself simplifies and is in fact independent of $\rho_c$,

$$i\{P_m, P_n\} = \frac{1}{8\pi G} \int d\phi\, e^{-i(m+n)\phi} (m-n) j = (m-n) J_{m+n}. \tag{5.31}$$

---

[25]It is straightforward to reintroduce the factors of $\ell$ into this result. The terms of cubic order in $(m, n)$ should be multiplied with $\ell^2$ whereas the linear terms are unchanged.

The other Poisson brackets are

$$i\{J_m, J_n\} = \frac{1}{8\pi G}\int d\phi\, e^{-i(m+n)\phi}(m-n)j\left(1 + \frac{\rho_c mn}{(1-\rho_c p)^2}\right),$$

$$i\{J_m, P_n\} = \frac{1}{8\pi G}\int d\phi\, e^{-i(m+n)\phi}\left(m^3 - \frac{\rho_c mj^2 - (m-n)p - \rho_c np^2 - \rho_c mn^2 p}{1-\rho_c p}\right). \tag{5.32}$$

In terms of these generators, the result with $\rho_c = 0$ becomes the $bms_3$ algebra in the limit of large AdS radius [72].

It is also interesting to consider the sub-AdS limit $\rho_c \to \infty$, which zooms in on a small patch of space. This gives

$$i\{Q_m, Q_n\} = \frac{m-n}{8\pi G}\int d\phi\, e^{-i(m+n)\phi}\frac{q^2}{q+\overline{q}},$$

$$i\{Q_m, \overline{Q}_n\} = \frac{n-m}{8\pi G}\int d\phi\, e^{-i(m-n)\phi}\frac{q\overline{q}}{q+\overline{q}}, \tag{5.33}$$

and their conjugates. In terms of $(J_m, P_m)$ we have

$$i\{P_m, P_n\} = i\{J_m, J_n\} = (m-n)J_{m+n},$$

$$i\{J_m, P_n\} = -nP_{m+n} + \frac{1}{8\pi G}\int d\phi\, e^{-i(m+n)\phi}\frac{mj^2}{p}. \tag{5.34}$$

# 6 Perturbation theory around global AdS$_3$ with finite cutoff

We now turn to the systematic treatment of quantizing the gravitational field perturbatively around global AdS$_3$ with a finite cutoff. We proceed by first identifying the phase space and computing the symplectic form and boundary charges. After simplifying the expressions using a field redefinition in Section 6.4, we promote functions on phase space to operators and define a Hilbert space in Section 7.

Global AdS$_3$ corresponds to (4.1) with $\mathcal{L} = \overline{\mathcal{L}} = -\frac{1}{4}$. We perform a simple coordinate redefinition in order to bring the metric at $\rho = \rho_c$ to our standard form $ds^2 = \frac{1}{\rho_c}dwd\overline{w}$. This procedure yields a particular case of (5.3),

$$ds^2 = \frac{d\rho^2}{4\rho^2} + \frac{1}{\rho}\frac{\left[(1-\rho\rho_c\mathcal{L}_0^2)dw + (\rho-\rho_c)\mathcal{L}_0 d\overline{w}\right]\left[(1-\rho\rho_c\mathcal{L}_0^2)d\overline{w} + (\rho-\rho_c)\mathcal{L}_0 dw\right]}{(1-\rho_c^2\mathcal{L}_0^2)^2}, \tag{6.1}$$

with

$$\mathcal{L}_0 = -\left(\frac{1-\sqrt{1+\rho_c}}{\rho_c}\right)^2. \tag{6.2}$$

For reference, in Appendix A we write out the six Killing vectors of global AdS$_3$ in the coordinates (6.1). It will sometimes be convenient to use the parameterization (see (1.4))

$$\rho_c = \alpha^2 - 1, \quad \mathcal{L}_0 = -\frac{1}{(1+\alpha)^2}. \tag{6.3}$$

## 6.1 Charge algebra near global AdS

We will start by expanding the Poisson bracket algebra of the charges (5.29) perturbatively around global AdS. To this end we write[26]

$$q(\phi) = -\frac{1}{2(1+\alpha)} + \frac{6}{c}\sum_n L_n e^{in\phi}, \quad \overline{q}(\phi) = -\frac{1}{2(1+\alpha)} + \frac{6}{c}\sum_n \overline{L}_n e^{-in\phi}.$$  (6.4)

Expanding in $1/c$ gives

$$
\begin{aligned}
i\{L_m, L_n\} &= \frac{c}{12\alpha}(m^3 - m)\delta_{m+n} + \frac{m-n}{4\alpha^2}\Big[(4 + 3\rho_c)L_{m+n} - \rho_c(2mn+1)\bar{L}_{-m-n}\Big] \\
&\quad + \frac{3\rho_c^2}{2\alpha^3 c}(m-n)\Big[(mn-1)(L^2)_{m+n} - (3mn+1)(\bar{L}^2)_{-m-n}\Big] \\
&\quad + \frac{3\rho_c}{\alpha^3 c}(m-n)(2 - \rho_c(mn-1))(L\bar{L})_{m+n} + \mathcal{O}(1/c^2), \\
i\{L_m, \bar{L}_n\} &= -\frac{\rho_c}{4\alpha^2}\Big[(m-n-2mn^2)L_{m-n} + (m-n+2m^2n)\bar{L}_{n-m}\Big] \\
&\quad - \frac{3\rho_c^2}{2c\alpha^3}\Big[(m-n-m^2n-3mn^2)(L^2)_{m-n} + (m-n+3m^2n+mn^2)(\bar{L}^2)_{n-m}\Big] \\
&\quad + \frac{3\rho_c}{c\alpha^3}(m-n)(2 - \rho_c(mn-1))(L\bar{L})_{m-n} + \mathcal{O}(1/c^2),
\end{aligned}
$$  (6.5)

where we have defined

$$(L^2)_m \equiv \sum_n L_n L_{m-n}, \qquad (L\bar{L})_m \equiv \sum_n L_{n+m}\bar{L}_n, \qquad (\bar{L}^2)_m \equiv \sum_n L_n L_{m-n}.$$  (6.6)

The result for $i\{\overline{L}_m, \overline{L}_n\}$ is obtained from $i\{L_m, L_n\}$ by interchanging barred and unbarred quantities. Taking $\rho_c \to 0$ gives back the usual pair of Virasoro algebras of asymptotically AdS$_3$ geometries.

## 6.2 Perturbation theory

Writing $w = \phi + it$ as usual, our goal is now to identify finite diffeomorphisms that preserve the form of the metric as $\rho = \rho_c$,

$$ds^2\Big|_{\rho_c} = \frac{d\rho^2}{4\rho_c^2} + \frac{dw\,d\overline{w}}{\rho_c} = \frac{d\rho^2}{4\rho_c^2} + \frac{dt^2 + d\phi^2}{\rho_c}.$$  (6.7)

Solving this problem in general appears to be difficult, and so we will proceed in perturbation theory. We work locally near the boundary circle at $\rho = \rho_c$ and $t = 0$, since this is sufficient for determining the boundary stress tensor, while the full extension into the bulk is pure gauge. As in Section 4.3, we look for a diffeomorphism $x^\mu = x'^\mu + \chi^\mu(\rho', t'\phi')$ with $\chi^\phi(\rho_c, 0, \phi') = A(\phi')$, and $\chi^t(\rho_c, 0, \phi') = B(\phi')$, where $(A(\phi'), B(\phi'))$ are freely specifiable periodic functions. Perturbation theory here means working order by order in powers of $(A, B)$. The full diffeomorphism is fixed by demanding $(\partial_t')^n \delta g_{\mu\nu}\big|_{t'=0, \rho'=\rho_c} = 0$, for $n = 0, 1, 2, \ldots$. The value of $n$ is correlated to the order in $(A, B)$ we work at.

---

[26]The $(L_n, \overline{L}_n)$ reduce to the usual Virasoro generators at $\rho_c = 0$; for nonzero $\rho_c$ they of course do not obey the Virasoro algebra.

We write the ansatz

$$\phi = \phi' + A(\phi') + \sum_{n=1}^{\infty} A_n(\phi') t'^n + (\rho' - \rho_c) \sum_{n=0}^{\infty} U_n(\phi') t'^n + \dots,$$

$$t = t' + B(\phi') + \sum_{n=1}^{\infty} B_n(\phi') t'^n + (\rho' - \rho_c) \sum_{n=0}^{\infty} V_n(\phi') t'^n + \dots,$$

$$\rho = \rho' + \rho' \sum_{n=0}^{\infty} C_n(\phi') t'^n + \rho'^2 \sum_{n=0}^{\infty} F_n(\phi') t'^n + \dots, \tag{6.8}$$

and then impose the boundary conditions to determine all unknown functions in terms of $(A(\phi'), B(\phi'))$, order by order in powers of the latter. This gives us a construction of the phase space, as labelled (modulo redundancies) by the freely specifiable periodic functions $(A(\phi'), B(\phi'))$.

For example, at lowest order in perturbation theory we find the nonzero functions

$$A_1 = -B', \quad B_1 = \frac{1}{\alpha^2} A', \quad C_0 = \frac{2}{\alpha} A', \quad U_0 = -\frac{1}{2\alpha} A'', \quad V_0 = \frac{1}{2\alpha} B''. \tag{6.9}$$

Implementing perturbation theory on the computer is straightforward. In particular, order by order one only encounters algebraic equations, allowing one to obtain the coordinate transformation to any desired order. We then transform the metric to the primed coordinates. The output of this procedure is a metric expressed in terms of $(A, B)$ and obeying the cutoff boundary conditions. We then compute the boundary stress tensor for this metric. Here we just write out the result to quadratic order

$$T_{tt} = \frac{c}{6} \left[ -\frac{1}{1+\alpha} - \frac{1}{\alpha} A' - \frac{1}{\alpha} A''' \right.$$

$$+ \frac{1}{\alpha^3} \left( A'A''' - B'B''' + \frac{3}{2} A''^2 - \frac{3}{2} B''^2 - \frac{1}{2} A'^2 + \frac{1}{2} B'^2 \right)$$

$$\left. + \frac{\rho_c}{\alpha^3} \left( -A''A'''' - A'''^2 + 2A'A''' - B'B''' + 2A''^2 - \frac{3}{2} B''^2 + \frac{1}{2} B'^2 \right) \right] + \dots,$$

$$T_{t\phi} = \frac{c}{6} \left[ -\frac{1}{\alpha} B' - \frac{1}{\alpha} B''' + \frac{1}{\alpha^3} \left( A'B''' + 3A''B'' - A'B' + A'''B' \right) \right.$$

$$\left. + \frac{\rho_c}{\alpha^3} \left( A''''B'' - A'''B' + A'''B''' - A'B' + A'B''' + A''B'' \right) \right] + \dots, \tag{6.10}$$

$T_{\phi\phi}$ is fixed by the trace relation (5.7). As a check, note that as we take the cutoff to infinity, corresponding to setting $\rho_c = 0$ and $\alpha = 1$, we recover the asymptotically AdS expressions (4.27).

The stress tensor in hand, we obtain the charges as usual from $Q[\xi] = \frac{i}{2\pi} \int_0^{2\pi} T_{ti} \xi^i d\phi$, the integration being performed at $t = 0$. Since we will use them later, we write out the Hamiltonian and momentum to cubic order,

$$H = -\frac{c}{6(1+\alpha)} + \frac{c}{24\pi} \int d\phi \left[ -\frac{1}{\alpha^3} (A + A'')'A' + \frac{1}{\alpha} (B + B'')'B' + \frac{\rho_c}{\alpha^5} A'^3 \right.$$

$$- \frac{(2 + 5\rho_c)}{\alpha^5} A'A''^2 + \frac{\rho_c}{\alpha^3} A'B'^2 + \frac{2(2-\rho_c)}{\alpha^3} A''B'B''$$

$$\left. + \frac{(2 + \rho_c)}{\alpha^3} A'B''^2 - \frac{2\rho_c}{\alpha^3} A''B''B''' \right] + \dots, \tag{6.11}$$

$$P = \frac{ic}{12\pi} \int d\phi \left[ -\frac{1}{\alpha}(A+A'')'B' + \frac{1+2\rho_c}{2\alpha^3}(A'^2)''B' - \frac{1}{2\alpha}(B'^2)''B' \right.$$
$$\left. + \frac{1}{\alpha^3}(A'B')''A' + \frac{\rho_c}{\alpha^3}(A''B'')''A' \right] + \dots \tag{6.12}$$

### 6.3 Computation of the symplectic form

We now wish to determine the symplectic form in terms of the functions $(A, B)$. The strategy is the same as in Section 4.2. We use the relation $i_{V_\xi}\Omega = -\delta Q[\xi]$ to compute $\Omega$ given expressions for the charges.

The first step is to obtain the variations $(\delta_\xi A, \delta_\xi B)$ under an infinitesimal (boundary condition preserving) diffeomorphism specified by vector field $\xi^\mu$. Recall the meaning of this statement. We start with some metric labelled by functions $(A, B)$. We then perform a transformation labelled by vector field $\xi^\mu$, using the construction in Section 5.2. We then label the resulting metric as $(A + \delta_\xi A, B + \delta_\xi B)$.

It is simplest to think in terms of composing the transformation (6.8) with a second infinitesimal transformation. Our goal here will be to work out $(\delta_\xi A, \delta_\xi B)$ to first order in $(A, B)$, in which case we can take the infinitesimal transformation to be

$$\phi' = \phi'' + \xi^\phi(\phi''),$$
$$t' = t'' + \xi^t(\phi''),$$
$$\rho' = \rho'' + \frac{2\rho_c}{\alpha}(\xi^\phi)' + \dots, \tag{6.13}$$

where we are restricting to the locus $(\rho'' = \rho_c, t'' = 0)$, which is all we need to compute the variation of the charges. Composing this with (6.8) and using $(\rho'' = \rho_c, t'' = 0)$ where convenient, we arrive at

$$\phi = \phi'' + A + \xi^\phi + A'\xi^\phi + A_1\xi^t + \frac{2\rho_c}{\alpha}(\xi^\phi)'U_0,$$
$$t = t'' + B + B'\xi^\phi + B_1\xi^t + \frac{2\rho_c}{\alpha}(\xi^\phi)'V_0, \tag{6.14}$$

where all functions on the right-hand side are functions of $\phi''$. We therefore have,

$$\delta_\xi A = \xi^\phi + A'\xi^\phi - B'\xi^t - \frac{\rho_c}{\alpha^2}A''(\xi^\phi)' + \dots,$$
$$\delta_\xi B = \xi^t + B'\xi^\phi + \frac{1}{\alpha^2}A'\xi^t + \frac{\rho_c}{\alpha^2}B''(\xi^\phi)' + \dots, \tag{6.15}$$

where the omitted terms are at least quadratic in $(A, B)$.

As a consistency check, we note that the relation $i_{V_\xi}\Omega = -\delta Q[\xi]$ and the antisymmetric nature of $\Omega$ imply the integrability condition

$$\delta_{\xi_1}Q[\xi_2] = -\delta_{\xi_2}Q[\xi_2], \tag{6.16}$$

where we use the notation $\delta_{\xi_1}Q[\xi_2] = i_{V_{\xi_1}}\delta Q[\xi_2]$. Using our expressions for the charges $Q[\xi]$ and the transformations (6.15) this can indeed be confirmed.

It is now straightforward to extract the symplectic form by solving $i_{V_\xi}\Omega = -\delta Q[\xi]$. The closure of $\Omega$ implies that we can write (at least in perturbation theory)

$$\Omega = \delta\Upsilon, \tag{6.17}$$

for some 1-form $\Upsilon$ on phase space. Writing out a general ansatz and then fixing coefficients one eventually arrives at

$$\Upsilon = \frac{ic}{12\pi}\int_0^{2\pi} d\phi \left[ -\frac{1}{\alpha}\left(A+A''\right)'\delta B + \frac{1+2\rho_c}{2\alpha^3}(A'^2)''\delta B - \frac{1}{2\alpha}(B'^2)''\delta B + \frac{1}{\alpha^3}(A'B')''\delta A \right.$$
$$\left. + \frac{\rho_c}{\alpha^3}(A''B'')''\delta A \right] + \dots .$$
(6.18)

Using integration by parts it is easy to see that upon taking the asymptotically AdS limit ($\rho_c = 0, \alpha = 1$) the result reduces to (4.28) up to exact forms, which do not contribute to $\Omega$.

From this expression we observe a six-dimensional degenerate subspace, associated to the six isometries of global AdS$_3$, which by definition have no effect on the metric and hence are pure gauge. Concretely, at lowest order we have

$$\Omega = \frac{ic}{12\pi}\int_0^{2\pi} d\phi \left[ -\frac{1}{\alpha}\left(\delta A + \delta A''\right)'\wedge \delta B \right] = \frac{ic}{12\pi}\int_0^{2\pi} d\phi \left[ \frac{1}{\alpha}\delta A \wedge \left(\delta B + \delta B''\right)' \right], \quad (6.19)$$

so we see that $\Omega$ has vanishing contraction against vector fields $V_\xi$ with $\delta_{V_\xi}A \sim e^{in\phi}$ or $\delta_{V_\xi}B \sim e^{in\phi}$ with $n = -1, 0, 1$. These linearized isometries obtain nonlinear corrections when we consider zero modes of $\Omega$ with more terms included.

To remove the degeneracy we can fix a gauge. The simplest option to write a mode expansion and simply omit the degenerate modes,

$$A(\phi) = \sum_{|n|>1} a_n e^{in\phi}, \quad B(\phi) = \sum_{|n|>1} b_n e^{in\phi}. \quad (6.20)$$

Substituting the mode expansion into $\Omega$ we thereby arrive at a non-degenerate symplectic form on the phase space with coordinates $(a_n, b_n)$ with $|n| > 1$.

## 6.4 Field redefinition

To facilitate quantization we now perform a field redefinition to Darboux-type coordinates in which the symplectic form has constant (i.e. field independent) components.

We wrote $\Upsilon$ in (6.18) in a form that motivates the following field redefinition

$$(A+A'')' = (\tilde{A}+\tilde{A}'')' + \frac{1+2\rho_c}{2\alpha^2}(\tilde{A}'^2)'' - \frac{1}{2}(\tilde{B}'^2)'',$$
$$(B+B'')' = (\tilde{B}+\tilde{B}'')' + \frac{1}{\alpha^2}(\tilde{A}'\tilde{B}')'' + \frac{\rho_c}{\alpha^2}(\tilde{A}''\tilde{B}'')'', \quad (6.21)$$

which yields

$$\Upsilon = \frac{ic}{12\pi}\int d\phi \left[ -\frac{1}{\alpha}\left(\tilde{A}+\tilde{A}''\right)'\delta\tilde{B} + (\text{quartic}) \right], \quad (6.22)$$

where "quartic" stands for terms like $A^3\delta B$ etc. More precisely, to interpret (6.21) we should use the mode expansion (6.20) along with the analogous expansion for the new fields,

$$\tilde{A}(\phi) = \sum_{|n|>1} \tilde{a}_n e^{in\phi}, \quad \tilde{B}(\phi) = \sum_{|n|>1} \tilde{b}_n e^{in\phi}, \quad (6.23)$$

so that (6.21) gives us a nonsingular relation between the old and new modes for $|n| > 1$.

We now apply this field redefinition to the Hamiltonian and momentum. The momentum is particularly simple:

$$P = -\frac{ic}{12\pi} \int d\phi \left[ \frac{1}{\alpha}(\tilde{A} + \tilde{A}'')'\tilde{B}' \right] + \text{quartic}. \tag{6.24}$$

The fact that the cubic terms are completely removed follows from the role of $P$ as the generator of $\phi$ translations; in the quantum theory it follows from momentum quantization as will become manifest below.

The Hamiltonian becomes

$$H = -\frac{c}{6(1+\alpha)} + \frac{c}{24\pi} \int d\phi \left[ -\frac{1}{\alpha^3}(\tilde{A} + \tilde{A}'')'\tilde{A}' + \frac{1}{\alpha}(\tilde{B} + \tilde{B}'')'\tilde{B}' + \frac{\rho_c}{\alpha^5}\tilde{A}'^3 + \frac{\rho_c}{\alpha^3}\tilde{A}'\tilde{B}'^2 \right.$$
$$\left. - \frac{\rho_c}{\alpha^5}\tilde{A}'\tilde{A}''^2 - \frac{2\rho_c}{\alpha^3}\tilde{A}''\tilde{B}'\tilde{B}'' + \frac{\rho_c}{\alpha^3}\tilde{A}'\tilde{B}''^2 \right] + \text{quartic}. \tag{6.25}$$

As expected, the Hamiltonian does retain non-quadratic terms, although we observe that these vanish upon setting $\rho_c = 0$.

For what follows we note that if we define

$$C = \frac{1}{\alpha}\tilde{A} + i\tilde{B}, \quad D = \frac{1}{\alpha}\tilde{A} - i\tilde{B}, \tag{6.26}$$

then

$$H = -\frac{c}{6(1+\alpha)} + \frac{c}{24\pi} \int d\phi \left[ -\frac{1}{2\alpha}(C + C'')'C' - \frac{1}{2\alpha}(D + D'')'D' \right.$$
$$\left. + \frac{\rho_c}{2\alpha^2}(C'^2 - C''^2)D' + \frac{\rho_c}{2\alpha^2}C'(D'^2 - D''^2) \right] + \text{quartic}. \tag{6.27}$$

# 7 Quantization

It is straightforward to quantize a theory given operators which obey the commutation relations of creation/annihilation operators. Given a quadratic symplectic form, like the one implies by (6.22), it is always possible to perform a linear field redefinition which produces the standard commutation relations of creation and annihilation operators. So towards this end, we first define new modes as

$$c_n = \sqrt{\frac{c|n|(n^2-1)}{12}} \left( \frac{1}{\alpha}\tilde{a}_n + i\tilde{b}_n \right),$$
$$c_n^\dagger = \sqrt{\frac{c|n|(n^2-1)}{12}} \left( \frac{1}{\alpha}\tilde{a}_{-n} + i\tilde{b}_{-n} \right),$$
$$d_n^\dagger = \sqrt{\frac{c|n|(n^2-1)}{12}} \left( \frac{1}{\alpha}\tilde{a}_n - i\tilde{b}_n \right),$$
$$d_n = \sqrt{\frac{c|n|(n^2-1)}{12}} \left( \frac{1}{\alpha}\tilde{a}_{-n} - i\tilde{b}_{-n} \right), \tag{7.1}$$

for $n > 1$. It will be useful, however, to define $c_{-n} = c_n^\dagger$ and $d_{-n} = d_n^\dagger$.[27] In the $\rho_c = 0$ limit the $c_n$ and $d_n$ modes will correspond to right and left movers. The symplectic form becomes

$$\Omega = -i \sum_{n>1} (\delta c_n^\dagger \wedge \delta c_n + \delta d_n^\dagger \wedge \delta d_n) \tag{7.2}$$

---

[27]Note that we have $(i\tilde{b}_n)^\dagger = i\tilde{b}_{-n}$ due to our choice of Euclidean time.

and from which, using the rule $[\cdot,\cdot]=i\{\cdot,\cdot\}$, we obtain the commutation relations

$$[c_m^\dagger,c_n]=[d_m^\dagger,d_n]=\delta_{m,n}\,.\tag{7.3}$$

The momentum is

$$P=\sum_{n>1}n(c_n^\dagger c_n-d_n^\dagger d_n)\,,\tag{7.4}$$

so that $c_n^\dagger$ creates $n$ positive units of momentum, and $d_n^\dagger$ creates $n$ negative units of momentum.

The Hamiltonian may now be written

$$H=H_0+H_2+H_3+\dots\,,\tag{7.5}$$

with

$$H_0=-\frac{c}{6(1+\alpha)}\,,\tag{7.6}$$

$$H_2=\sum_{n>1}\frac{n}{\alpha}\left[c_n^\dagger c_n+d_n^\dagger d_n\right]\,,\tag{7.7}$$

and

$$H_3=\frac{\sqrt{3}i\rho_c}{\sqrt{c}\alpha^2}\sum_{m,n,p}A_{m,n,p}(c_m c_n d_p-c_p d_m d_n)\delta_{p,m+n}\,,\tag{7.8}$$

where we defined

$$A_{m,n,p}\equiv\begin{cases}0 & \text{if }m,n,p\in\{-1,0,1\}\,,\\ \operatorname{sgn}(mnp)\sqrt{\dfrac{|mnp|}{(m^2-1)(n^2-1)(p^2-1)}}(mn+1) & \text{otherwise}\end{cases}\tag{7.9}$$

We also note that the general charges take the following form to linear order,

$$Q_n=-\frac{c}{12(1+\alpha)}\delta_{n,0}+i\sqrt{\frac{c}{12}}\operatorname{sgn}(n)\sqrt{|n|(n^2-1)}c_n+\text{quadratic}\,,$$

$$\overline{Q}_n=-\frac{c}{12(1+\alpha)}\delta_{n,0}-i\sqrt{\frac{c}{12}}\operatorname{sgn}(n)\sqrt{|n|(n^2-1)}d_n+\text{quadratic}\,.\tag{7.10}$$

## 7.1 Spectrum

We can now work out the spectrum to the first few orders in the $1/\sqrt{c}$ expansion. Starting with the free theory, $H=H_0+H_2$, we define the obvious vacuum state $c_n|0\rangle=d_n|0\rangle=0$, for $n=2,3\dots$, and then build up the Hilbert space by acting with the creation operators $(c_n^\dagger,d_n^\dagger)$. This is of course the theory of a free scalar, except that the $n=-1,0,1$ modes are absent. In term of the number operators $N=\sum_{n>1}nc_n^\dagger c_n$ and $\overline{N}=\sum_{n>1}nd_n^\dagger d_n$ we have

$$E=-\frac{c}{6(1+\alpha)}+\frac{N+\overline{N}}{\alpha}+\dots\,,$$

$$P=N-\overline{N}\,.\tag{7.11}$$

Next we go to $\mathcal{O}(1/\sqrt{c})$ by including $H_3$. The spectrum (7.11) has degeneracies, and so we need to apply degenerate perturbation theory, which as usual involves computing the matrix

elements of $H_3$ between states within the same degenerate subspace and then diagonalizing within each subspace. Since $[P, H_3] = 0$, the only potentially nonzero matrix elements are those involving states of the same $P$. Along with the equal energy requirement, we see that the two states must have the same $N$ and the same $\overline{N}$ values. We now note from the explicit form of $H_3$ that each term involves either a single rightmoving $c$-type operator, or a single leftmoving $d$-type operator. Such operators necessarily change the value of either $N$ or $\overline{N}$, and hence the matrix elements in question all vanish. The vanishing of the matrix elements of $H_3$ between degenerate states implies that the energy is uncorrected at $\mathcal{O}(1/\sqrt{c})$, as predicted from the $T\overline{T}$ analysis.

Given that $H_3$ does not affect the spectrum we can expect to be able to remove it by a unitary transformation. Indeed, if we define

$$U = e^{\frac{i}{\sqrt{c}} K}, \tag{7.12}$$

with (the primed sum indicates that the $p = 0$ term should be omitted)

$$K = -\frac{\sqrt{3}\rho_c}{2\alpha} {\sum_{m,n,p}}' \frac{1}{p} A_{m,n,p}(c_m c_n d_p - c_p d_m d_n)\delta_{p,m+n} \tag{7.13}$$

it is easy to see that

$$H_0 + H_2 + H_3 = U(H_0 + H_2)U^\dagger + \text{quartic}, \tag{7.14}$$

where the quartic terms are $\mathcal{O}(1/c)$. The relation (7.14) makes it manifest that there are no corrections to the energy spectrum at $\mathcal{O}(1/\sqrt{c})$. We do of course expect nontrivial corrections to the spectrum at $\mathcal{O}(1/c)$ coming from $H_4$, which we have not computed.

## 8 Expectation for the spectrum at $\mathcal{O}(1/c)$ and beyond

A main result of this paper was the classical Poisson bracket algebra obeyed by the boundary charges $(Q_n, \overline{Q}_n)$. The quantum version of this algebra should act on the Hilbert space and is expected to determine the spectrum. Due to the nonlinear nature of the algebra there are severe ordering ambiguities in passing from the classical to quantum case. One way to resolve these is to extend our analysis in Section 7 to higher orders in the $1/c$ expansion, which here plays the role of $\hbar$. Rather than doing so, in this section we will see how far we can get in deriving the spectrum by making some educated guesses regarding how the charge algebra acts in the quantum theory. As we will see, with some plausible assumptions we can obtain the $1/c$ corrections to the energies, as well as get a glimpse of how things work at higher orders. A more systematic treatment is left to the future.

We introduce generators $(L_n, \overline{L}_n)$ as in (6.4). The energy and momentum operators are

$$H = -\frac{c}{6(1+\alpha)} + L_0 + \overline{L}_0, \quad P = L_0 - \overline{L}_0. \tag{8.1}$$

Now, with $\rho_c = 0$, we have a pair of Virasoro algebras and $(L_{-n}, \overline{L}_{-n})$ act as ladder operators for $(H, P)$, yielding the spectrum $H = -\frac{c}{12} + N + \overline{N}$ and $P = N - \overline{N}$. This ladder property is no longer true at nonzero $\rho_c$, as we see from (6.5). We therefore define

$$L'_n = \frac{\alpha+1}{2}L_n - \frac{\alpha-1}{2}\overline{L}_{-n},$$
$$\overline{L}'_n = \frac{\alpha+1}{2}\overline{L}_n - \frac{\alpha-1}{2}L_{-n}, \tag{8.2}$$

which obey

$$[H, L'_n] = -\frac{n}{\alpha} L'_n + \mathcal{O}(1/c),$$
$$[H, \overline{L}'_n] = -\frac{n}{\alpha} \overline{L}'_n + \mathcal{O}(1/c),$$
$$[P, L'_n] = -nL'_n,$$
$$[P, \overline{L}'_n] = n\overline{L}'_n, \tag{8.3}$$

where we have replaced Poisson brackets by commutators using the rule $i\{\cdot, \cdot\} = [\cdot, \cdot]$. The form of the primed generators may be understood by comparing to the Killing vectors of global AdS$_3$ written in (A.7).

At order $c^0$ we define the vacuum state to obey $L'_n |0\rangle = \overline{L}'_n |0\rangle = 0$ for $n = -1, 0, 1$. We then fill out the Hilbert space by acting with strings of $L'_{-n}$ and $\overline{L}'_{-n}$ operators, and let $(N, \overline{N})$ be the associated level numbers. From (8.3) this gives the energy spectrum

$$E = -\frac{c}{6(1+\alpha)} + \frac{N + \overline{N}}{\alpha} + \mathcal{O}(1/c), \tag{8.4}$$

in agreement with (1.2). We also have $P = N - \overline{N}$.

Now we go to order $1/c$. At this order we find

$$[H, L'_n] = -\frac{n}{\alpha} L'_n - \frac{12\rho_c}{c\alpha^2} n(L'\overline{L}')_n,$$
$$[H, \overline{L}'_n] = -\frac{n}{\alpha} \overline{L}'_n - \frac{12\rho_c}{c\alpha^2} n(L'\overline{L}')_{-n}, \tag{8.5}$$

where are writing $(L'\overline{L}')_n = \sum_p L'_{n+p} \overline{L}'_p$, and we note that this operator suffers from an ordering ambiguity at this order. This leads us to look for a modified Hamiltonian, for which $(L'_n, \overline{L}'_n)$ act as ladder operators. We define

$$H' = H - \Delta H, \quad \Delta H = \frac{12\rho_c}{c\alpha} L_0 \overline{L}_0 + \dots. \tag{8.6}$$

The ... refer to nonzero mode contributions to $\Delta H$, which we return to in a moment. We then compute

$$H' |\psi_{N,\overline{N}}\rangle = \left[ -\frac{c}{6(1+\alpha)} + \frac{(N + \overline{N})}{\alpha} \right] |\psi_{N,\overline{N}}\rangle + \mathcal{O}(1/c^2), \tag{8.7}$$

where $|\psi_{N,\overline{N}}\rangle$ denotes a state with levels $(N, \overline{N})$, produced by acting on the vacuum with a string of $L'_{-n}$ and $\overline{L}'_{-n}$ operators. To obtain (8.7) we first of all chose to define $(L'\overline{L}')_n$ via a symmetric ordering. Second, we have only explicitly verified that the part of (8.7) which is sensitive to the zero mode part of $\Delta H$ is satisfied, and we have assumed that the nonzero mode part of $\Delta H$ can be chosen to satisfy the rest of (8.7). This latter point is a gap in our argument that needs to be filled.

Assuming this holds, we have now succeeded in diagonalizing $H'$. However, we interested in the eigenvalues of $H$, not $H'$. To obtain the former, we write $H = H' + \Delta H$, and view $\Delta H$ as a perturbation of $H'$, whose eigenvectors and eigenvalues we know. The $1/c$ correction to the eigenvalues of $H$ are given by standard first order perturbation theory, namely by evaluating the expectation value of the perturbation in the unperturbed state. To implement this we write

$$\Delta H = \frac{3\rho_c}{c\alpha} \left( (L_0 + \overline{L}_0)^2 - (L_0 - \overline{L}_0)^2 \right). \tag{8.8}$$

Using (8.5) we obtain

$$\langle \psi_{N,\overline{N}}|\Delta H|\psi_{N,\overline{N}}\rangle = \frac{3\rho_c}{c\alpha}\left[\frac{(N+\overline{N})^2}{\alpha^2} - (N-\overline{N})^2\right] + \mathcal{O}(1/c^2). \tag{8.9}$$

Combining this with the lower contributions we arrive at

$$E = -\frac{c}{6(1+\alpha)} + \frac{N+\overline{N}}{\alpha} + \frac{3\rho_c}{c}\frac{(N+\overline{N})^2 - \alpha^2(N-\overline{N})^2}{\alpha^3} + \mathcal{O}(1/c^2), \tag{8.10}$$

in agreement with (1.2).

We can also shed some light on how the all orders $T\overline{T}$ energy spectrum will arise. We note that the exact spectrum,

$$E = \frac{c}{6\rho_c}\left(1 - \sqrt{1 + \rho_c - \frac{12}{c}\rho_c(N+\overline{N}) + \frac{36}{c^2}\rho_c^2(N-\overline{N})^2}\right) \tag{8.11}$$

can be rewritten as

$$E - \frac{3\rho_c}{c}(E^2 - P^2) = -\frac{c}{12} + N + \overline{N}, \tag{8.12}$$

with $P = N - \overline{N}$. This indicates that if we work out $(H, P)$ expressed in terms of $(c_n, d_n)$ obeying $[c_m^\dagger, c_n] = [d_m^\dagger, d_n] = \delta_{m,n}$ and apply a unitary transformation we will obtain

$$U\left(H - \frac{3\rho_c}{c}(H^2 - P^2)\right)U^\dagger = -\frac{c}{12} + \sum_{n>1} n(c_n^\dagger c_n + d_n^\dagger d_n). \tag{8.13}$$

Indeed we already established a low order version of this in (7.14) when we transformed away the cubic terms in $H$ by a unitary transformation. This line of thought is similar to [40], where the interpretation of the $T\overline{T}$ deformation as implementing a unitary transformation was developed.

## 9 Discussion

In this work we developed the canonical formulation of pure 3D gravity with Dirichlet boundary conditions imposed on a timelike cylinder of finite spatial circumference. We computed the Poisson bracket algebra of observables in this theory, which are the Fourier modes of the boundary stress tensor, and obtained a nonlinear algebra. This nonlinear algebra is a one-parameter deformation of the usual pair of Virasoro algebras present with asymptotically AdS$_3$ boundary conditions.

We initiated quantization of this system by applying a strategy analogous to that used in the coadjoint orbit method. In particular, we restricted attention to the space of solutions connected to global AdS by boundary condition preserving diffeomorphisms, and used the diffeomorphism functions as coordinates on the orbit. Unlike in the asymptotically AdS$_3$ case where the stress tensor is readily expressed in terms of these functions via the Schwarzian derivative, at finite $\rho_c$ no analogous parameterization is immediately apparent, and so we worked perturbatively by expanding the diffeomorphisms around the identity. We carried this out far enough to check that the free spectrum and the leading cubic interaction are in agreement with the prediction from $T\overline{T}$ (in particular the cubic interaction was shown to vanish after implementing a unitary transformation).

The charges considered here are different functions on the phase space than those in [44], and their algebra differs as well. For the purposes of quantizing the theory, which was anticipated to lead to problems in the setup of [44], our choice to label the charges in terms of fixed functions of the state-independent coordinates on a fixed time surface turned out to be well-suited. It would be interesting to apply this procedure for general $T\overline{T}$-deformed theories within a purely field-theoretical framework.

The most obvious extension of this work is to carry out quantization to higher orders in the $1/c$ expansion. This is quite challenging if one follows the precise method used here, but is likely simplified by performing a field redefinition at an appropriate stage. We expect to be able to reproduce the full $T\overline{T}$ spectrum, inasmuch as the QFT derivation of this spectrum only involves assuming certain properties of the stress tensor which appear to be satisfied by the Dirichlet cutoff prescription. Of course, we would like to obtain more than just the energy spectrum; in particular we aim for quantum expressions for the stress tensor operator, which would allow us to compute its correlators at finite $\rho_c$, which in turn would shed light on the (non)locality of $T\overline{T}$-deformed CFTs.

There are also other natural extensions of this work, such as the quantization of other orbits describing conical defects (see [79, 80]) and black holes, and considering curved boundaries (e.g. [81, 82]), which would make contact with the challenge of defining the $T\overline{T}$ deformation on a curved background geometry [83]. Finally, it would be interesting to apply this formalism to different numbers of dimensions, most immediately to two-dimensional JT gravity where one could make the connection with the results of [30, 31].

## Acknowledgements

We thank Eric D'Hoker and Monica Guica for useful discussions. P.K. and R.M. are supported in part by the National Science Foundation under grant PHY-1914412.

## A   Global AdS$_3$ Killing vectors

Here we write out the form of the six Killing vectors of global AdS$_3$, adapted to our coordinates with a boundary at $\rho = \rho_c$. The metric takes the form

$$ds^2 = \frac{d\rho^2}{4\rho^2} + \frac{1}{\rho} \frac{\left[(1-\rho\rho_c\mathcal{L}\overline{\mathcal{L}})dw + (\rho-\rho_c)\overline{\mathcal{L}}d\overline{w}\right]\left[(1-\rho\rho_c\mathcal{L}\overline{\mathcal{L}})d\overline{w} + (\rho-\rho_c)\mathcal{L}dw\right]}{(1-\rho_c^2\mathcal{L}\overline{\mathcal{L}})^2}, \quad \text{(A.1)}$$

with

$$\mathcal{L} = \overline{\mathcal{L}} = \mathcal{L}_0 = -\left(\frac{1-\sqrt{1+\rho_c}}{\rho_c}\right)^2 \quad \text{(A.2)}$$

and we are taking $w = \phi + it$ and $\overline{w} = \phi - it$ with $\phi \cong \phi + 2\pi$. In these coordinates the origin is at $\rho = -1/\mathcal{L}_0$.

On the other hand, a more standard form of the global AdS$_3$, adapted to the asymptotic boundary is

$$ds^2 = \frac{d\rho'^2}{4\rho'^2} + \frac{1}{\rho'}\left(1+\frac{\rho'}{4}\right)^2 dt'^2 + \frac{1}{\rho'}\left(1-\frac{\rho'}{4}\right)^2 d\phi'^2, \quad \text{(A.3)}$$

with $\phi' \cong \phi' + 2\pi$. The two forms of the metric are related by

$$\rho' = -4\mathcal{L}_0\rho, \quad t' = -\frac{t}{\alpha}, \quad \phi' = \phi, \quad \text{(A.4)}$$

with $\alpha = \sqrt{1 + \rho_c}$.

Writing $w' = \phi' + it'$ the Killing vectors in the primed coordinates system are $\partial_{w'}$, $\partial_{\overline{w'}}$ and

$$\xi_n = e^{inw'}\left[\frac{1 + \frac{\rho'^2}{16}}{1 - \frac{\rho'^2}{16}}\partial_{w'} + \frac{1}{2}\frac{\rho'}{1 - \frac{\rho'^2}{16}}\partial_{\overline{w'}} + in\rho'\partial_{\rho'}\right], \quad n = \pm 1,$$

$$\overline{\xi}_n = e^{-in\overline{w'}}\left[\frac{1 + \frac{\rho'^2}{16}}{1 - \frac{\rho'^2}{16}}\partial_{\overline{w'}} + \frac{1}{2}\frac{\rho'}{1 - \frac{\rho'^2}{16}}\partial_{w'} - in\rho'\partial_{\rho'}\right], \quad n = \pm 1. \tag{A.5}$$

These obey

$$[\xi_1, \xi_{-1}] = -2i\partial_{w'}, \quad [\overline{\xi}_1, \overline{\xi}_{-1}] = 2i\partial_{\overline{w'}}. \tag{A.6}$$

The Killing vectors in the unprimed coordinates are then obtained from (A.4); here we just write their form when restricted to the $\rho = \rho_c$ surface, and evaluated at $t = 0$,

$$\xi_n = e^{in\phi}\left[\frac{\alpha + 1}{2}\partial_w + \frac{\alpha - 1}{2}\partial_{\overline{w}}\right],$$

$$\overline{\xi}_n = e^{-in\phi}\left[\frac{\alpha + 1}{2}\partial_{\overline{w}} + \frac{\alpha - 1}{2}\partial_w\right]. \tag{A.7}$$

# B  Gravitational boundary charges

In this appendix we include a few more details on the construction of boundary charges within the covariant phase space approach to gravity. The general formalism is discussed in many places (for pedagogical treatments relevant to our considerations we recommend [27,72]); in practice, almost everything we need is contained in [54].

We consider the Einstein-Hilbert action in $(d+1)$-dimensions, with Lagrangian $(d+1)$-form

$$L = -\frac{1}{16\pi G}(R - 2\Lambda)\sqrt{g}\,d^{d+1}x. \tag{B.1}$$

Writing its variation as $\delta L = E^{\mu\nu}\delta g_{\mu\nu} + d\Theta$ yields the symplectic potential $d$-form

$$\Theta = -\frac{1}{16\pi G}\left(g^{\mu\alpha}\nabla^\nu\delta g_{\alpha\nu} - g^{\alpha\beta}\nabla^\mu\delta g_{\alpha\beta}\right)\sqrt{g}(d^d x)_\mu. \tag{B.2}$$

Using

$$\nabla_\mu\delta g_{\alpha\beta} = g_{\nu\alpha}\delta\Gamma^\nu_{\mu\beta} + g_{\nu\beta}\delta\Gamma^\nu_{\alpha\mu} = 2g_{\nu(\alpha}\delta\Gamma^\nu_{\beta)\mu}, \tag{B.3}$$

we can rewrite $\Theta$ as

$$\Theta = -\frac{1}{16\pi G}\left(g^{\alpha\beta}\delta\Gamma^\mu_{\alpha\beta} - g^{\mu\alpha}\delta\Gamma^\beta_{\alpha\beta}\right)\sqrt{g}(d^d x)_\mu. \tag{B.4}$$

Viewing $\Theta$ as a 1-form on phase space and applying the exterior derivative $\delta$ it is straightforward to compute $\delta\Theta = J^\alpha\sqrt{g}(d^d x)_\alpha$ with

$$J^\alpha = \frac{1}{16\pi G}\left[\delta\Gamma^\alpha_{\mu\nu} \wedge \left(\delta g^{\mu\nu} + \frac{1}{2}g^{\mu\nu}\delta\ln g\right) - \delta\Gamma^\nu_{\mu\nu} \wedge \left(\delta g^{\alpha\mu} + \frac{1}{2}g^{\mu\alpha}\delta\ln g\right)\right], \tag{B.5}$$

which (up to normalization) is the result in [54]. The symplectic form is then obtained by integrating over a Cauchy slice, $\Omega = i\int_\Sigma d\Sigma_\alpha\sqrt{g}J^\alpha$, where the factor of $i$ is due to our choice of Euclidean signature.

The other main result we need is (3.16)-(3.17), which establishes that the charges associated to diffeomorphisms are pure boundary terms. This is a straightforward, though moderately lengthy, computation. The result for $X^{\alpha\nu}$ is given in [54]. Our expression in (3.17) has some sign differences compared to [54], which is due to the fact that in our conventions we take $\delta_\xi g_{\mu\nu}$ to commute with $\delta$.

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
