# Peer review of "D Gravity in a Box"

_SciPost Physics, doi:SciPost Phys. 11, 070 (2021)_

## Round 2 · Referee Report · Anonymous · 2021-8-21

Strengths
1-Makes very clear what the key findings are. The paper is self-contained and pedagogical.
2-Explains the origin of "state-dependent conformal symmetry", which was always a puzzling aspect of the literature on $T\bar{T}$.
3-The paper derives the Alekseev-Shatashvili symplectic form by working in metric variables.
4-Concrete, detailed computations that pass consistency checks.
Weaknesses
1-The magic of the solvability of the $T\bar{T}$ spectrum in field theory remains unexplained from the gravitational description.
Report
This paper is an important contribution to the holographic investigations of $T\bar{T}$-deformed CFTs. It resolves a major confusion in the literature, where a structure appearing in these theories was interpreted as "state-dependent conformal symmetry". The paper investigates this question systematically, and explains that the corresponding "charges" are not conserved, instead they are just some nice set of observables obeying a nontrivial rigid algebra. I expect that the formalism presented in this paper will be the starting point of any future investigation in this direction. The paper is very well-written and pedagogical, hence I recommend it for publication.
I have one question to the authors: Is the formalism adaptable to JT gravity? Could the program of the paper be carried out to all orders in this simpler theory? JT gravity has a two-dimensional phase space, and in [31] very suggestive results were obtained about the relation between the cutoff JT theory and the $T\bar{T}$-deformed Schwarzian theory.
Requested changes
1-I found a couple of typos:
Both in (3.12) and above (B.5) the authors write $d^dx$ instead of $dx^d$. In sec. 6.4 they write "we now which to perform"
Author: Ruben Monten on 2021-08-30 [id 1719]
(in reply to Report 1 on 2021-08-21)We thank the referee for the very interesting suggestion and for pointing out the typos.
Regarding the typos: in the new version we have fixed the definition of the hypersurface volume form in (3.13) so that (3.12) and (B.5) are consistent with it.
As for the suggestion to apply this formalism to JT gravity at finite cutoff: indeed, we believe that it should apply. Moreover, given the explicit solutions in Section 2 of [31], it is conceivable that this program can be carried out to all orders in perturbation theory. We added a comment to this effect in the discussion section and hope to report on the results in a separate article in the future.

---

## Round 3 · Author Response

This is a resubmission, corresponding to v3 on the arXiv, addressing the referee's comments.

---

## Round 3 · List of Changes

• We added a comment in the discussion session about the analog formalism for JT gravity, as suggested by the referee.
  • We corrected the typos and changed the definition of the hypersurface volume form in (3.13) so that (3.12) and (B.5) are consistent with it.

---

## Editorial Decision

published